# Phonons as a platform for non-Abelian braiding and its manifestation in layered silicates

Bo Peng [1,4✉], Adrien Bouhon [2,4✉], Bartomeu Monserrat [1,3,4✉] & Robert-Jan Slager [1,4✉]

Topological phases of matter have revolutionised the fundamental understanding of band theory and hold great promise for next-generation technologies such as low-power electronics or quantum computers. Single-gap topologies have been extensively explored, and a large number of materials have been theoretically proposed and experimentally observed. These ideas have recently been extended to multi-gap topologies with band nodes that carry non-Abelian charges, characterised by invariants that arise by the momentum space braiding of such nodes. However, the constraints placed by the Fermi-Dirac distribution to electronic systems have so far prevented the experimental observation of multi-gap topologies in real materials. Here, we show that multi-gap topologies and the accompanying phase transitions driven by braiding processes can be readily observed in the bosonic phonon spectra of known monolayer silicates. The associated braiding process can be controlled by means of an electric field and epitaxial strain, and involves, for the first time, more than three bands. Finally, we propose that the band inversion processes at the $\Gamma$ point can be tracked by following the evolution of the Raman spectrum, providing a clear signature for the experimental verification of the band inversion accompanied by the braiding process.

[1] TCM Group, Cavendish Laboratory, University of Cambridge, J. J. Thomson Avenue, Cambridge CB3 0HE, United Kingdom. [2] Nordic Institute for Theoretical Physics (Nordita), Stockholm University and KTH Royal Institute of Technology, Hannes Alfvéns väg 12, Stockholm SE-106 91, Sweden. [3] Department of Materials Science and Metallurgy, University of Cambridge, 27 Charles Babbage Road, Cambridge CB3 0FS, United Kingdom. [4] These authors contributed equally: Bo Peng, Adrien Bouhon, Bartomeu Monserrat, Robert-Jan Slager. ✉email: bp432@cam.ac.uk; adrien.bouhon@su.se; bm418@cam.ac.uk; rjs269@cam.ac.uk

Following the discovery of topological insulators in nearly free electronic systems[1,2], the past decade has seen a surge of interest into topological phenomena in band insulators and metals[3]. Combining topology with space group symmetries has resulted in a myriad of topological characterisations of materials. In this context, a rather universal framework has been established to study single-gap topology: the determination of topologically inequivalent band structures using the band representations at high-symmetry points in the Brillouin zone[4–6], an approach that matches the full K-theory result in some scenarios[4,7]. These momentum space constraints have subsequently been compared to real space constraints, resulting in versatile classification schemes to characterise topological materials by identifying which of these combinations have an atomic limit[8,9].

Recent work is uncovering new physics beyond these symmetry indicated schemes that depends on *multi-gap* conditions, especially in systems with $C_2T$ or $PT$ symmetries. A system with these symmetries can be described using a real Hamiltonian, where band nodes at different gaps carry non-Abelian charges, typically called frame charges[10–15]. As a result, the momentum space braiding of a node in one gap around a node in an adjacent gap can change their charge, leading to nodes with same-valued charges in a specific gap. This creates an obstruction to annihilate such nodes that can be characterised with a new invariant, known as Euler class, that is computed over patches in the Brillouin zone that contain all the nodes of that two-band subspace, i.e. the two bands around the gap hosting the (possibly multiple) pairs of stable nodes. Generically, an Euler class $\chi$ indicates the presence of $2|\chi|$ stable nodes (i.e. with the same charge) within the two-band space over that Brillouin zone patch. These stable nodes can be annihilated only by the inverse braiding process, thus necessitating extra bands. As these inverse braiding processes can be achieved by trivial bands, they can formally be seen as a new form of fragile topology[16–21]. However, this fragile topology is fundamentally distinct to symmetry indicated fragile topology: rather than a subset of bands that can be trivialised by simply adding extra bands without any notion of charge conversion processes, the extra bands form an essential part in the description of the Euler class.

Interest is growing in the study of multi-gap topologies[22–27], with recent developments including new dynamical quench signatures[28] and a very recent realisation in an acoustic metamaterial[15]. Despite these advances, the multi-gap condition is complicated by the Fermi-Dirac distribution of electrons in materials, and as a result multi-gap topology has not yet been observed in real materials.

We propose that phonons, which are a bosonic excitation not subject to Fermi-Dirac statistics, provide a viable platform to observe multi-gap topology. We identify a monolayer silicate as a candidate material, and show that an electric field and epitaxial strain can be used to induce band inversions which are accompanied by the transfer of nodes between adjacent gaps, thereby transferring non-Abelian frame charges and non-trivial patch Euler class. This is achieved through the symmetry-constrained braiding of band nodes, resulting in a multi-gap topological phase. Different from the braiding in the metamaterial[15], we can realise many-band (more than three) braiding processes in phonons of monolayer silicate. We further show that the evolution of Raman peaks can be used to track the band inversions and the accompanying braiding process, providing a clear experimental signature to identify multi-gap topology in real materials. Interest in the topological features of phonon bands has recently grown[29–40], but for single-gap topology, phonons have traditionally received less attention than electrons. The bosonic nature of phonons should make them the prime platform for the study of multi-gap topology.

## Results

**Silicates.** Layered silicates are ubiquitous in soils and minerals throughout the world[41,42]. The surface layer of silicates consists of a silicon-centred tetrahedron, with the oxygen atoms forming a coplanar hexagonal Kagome lattice, as shown in Fig. 1a. Physical models based on the two-dimensional (2D) Kagome lattice exhibit rich phenomenology, from Dirac fermions to flat bands, and as a result there is much interest in this structural pattern. The experimental realisation of monolayer Kagome silicates can be dated back to the 1990s[43], but more recently various synthesis strategies have been developed to obtain 2D Kagome silicate or silica on multiple substrates[44–48]. These developments enable researchers to study the structural[49–55], vibrational[56–58], electronic[59–61], mechanical[62,63], and chemical properties[64] of this material family and to explore the various physical phenomena associated with 2D Kagome lattices.

**Phonon band structure.** The monolayer Kagome silicate $Si_2O_3$ crystallises in the $P6mm$ space group (No. 183). Different from free standing bilayer silica, monolayer silicate $Si_2O_3$ is grown on substrates, as shown in Fig. 1a. Using passivating hydrogen atoms to mimic the substrates, we compute the phonon dispersion of monolayer $Si_2O_3$ from first principles (see Supplementary Fig. 1 for the full phonon dispersion). The resulting phonon dispersion has three groups of Kagome flat bands between 18 and 36 THz (Fig. 1b). Group 1 (orange region) contains typical Kagome bands with two Dirac bands capped by another flat band, which belong to bands 10–12 in the full phonon dispersion. Group 2 (green region) consists of four bands (bands 13–16), with two flat bands capping the Dirac bands in the middle. Group 3 (navy region) contains bands 17–19, and the flat band is below the Dirac bands. In the following, we number the bands $\{\mathcal{B}_n\}_{n=1,\ldots,21}$ from lower

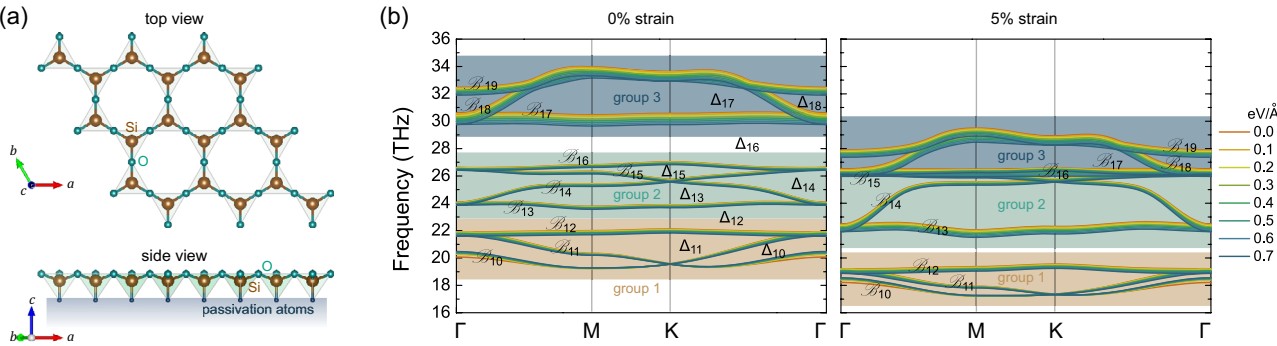

**Fig. 1 Phonons in monolayer Kagome silicate $Si_2O_3$. a** Crystal structure of monolayer Kagome silicate $Si_2O_3$. **b** Kagome bands of the phonon spectra of 0 and 5% strained silicate under different electric fields.

to higher frequencies, and we label the partial gaps between each two successive bands $(\mathcal{B}_n, \mathcal{B}_{n+1})$ as $\{\Delta_n\}_{n=1,\dots,20}$ with frequencies $E_n \leq \Delta_n \leq E_{n+1}$ (see Fig. 1).

**Electric field induced band inversions and braiding.** The key discovery of our work is that it is possible to controllably braid Kagome band nodes in monolayer $Si_2O_3$ using strain and/or an external electric field. Interestingly, we find two types of braiding. The first type involves the three bands in group 1 (orange region) in Fig. 1(b), and represents a realistic proposal for the first material realisation of a multi-gap phase. The second type of braiding involves more than three bands, including groups 2 and 3, shown in the green and navy regions in Fig. 1b, and represents the first proposal of a system that can host these many-band (more than three) braiding processes.

In the generic braiding process for three bands[10–15,28], the frame charges $q$ associated with the band nodes take values in the conjugacy classes $\pm q$ of the quaternion group $\mathbb{Q}$, where $q \in \{1, i, j, k\}$ satisfying $i^2 = j^2 = k^2 = -1$, $ij = k$, $jk = i$ and $ki = j$. Specifically, $\pm i$ and $\pm j$ characterise single nodes in each of the respective gaps (first and second), whereas $\pm k$ represents a pair of nodes in each gap. Finally, $-1$ captures the stability of two nodes of the same gap[10]. These charges can be manipulated through the braiding process of band nodes of different gaps in momentum space to arrive at configurations where a specific gap features the same charges. This results in an obstruction to annihilation that can be quantified via a non-trivial Euler invariant over a patch in the Brillouin zone that contains all the nodes of this band subspace[13].

As an extension to the three-band setup characterised by the quaternion group, when more bands are present the above ideas must be generalised to the so-called Salingaros' vee groups[10,13,14] (see details in the Method section). The main property of interest for our braiding processes is that nodes of adjacent gaps reproduce the three-band case and have anti-commuting charges, whereas charges of non-neighbouring gaps commute. In particular, any stable simple node in a gap is characterised by the charge $q$ that takes, as before, $\pm q$ values, while the charge $q^2 = -1$ indicates a stable pair of nodes. In other words, these configurations pose an intricate extension of the three-band case[10], featuring opposite kinds of each charge and a charge of $-1$ in each gap. The regions where several adjacent gaps $(\Delta_n, \dots, \Delta_{n+M})$ are each hosting a simple node can be characterised through products of non-Abelian charges $(q_n \cdots q_{n+M})$[10].

Importantly, in our context the configurations of the nodes are constrained by the crystalline symmetries of the system, leading to a projection of the braid trajectories on the different high-symmetry points and lines of the Brillouin zone (see 'Methods'). The band inversions at these momenta thus readily indicate the braiding processes. We corroborate the non-Abelian nature of the band inversions through the direct computation from the first-principle data of the patch Euler classes and the non-Abelian charges of the nodes transferred across adjacent gaps.

The band order of the Kagome bands can be inverted at different strains and under different electric fields. We calculate the strain-dependent (from $-2$ to 8%) phonon dispersion under electric fields from $-0.9$ to 2.0 eV/Å. No imaginary modes are observed even for 7% strained $Si_2O_3$ under an electric field of 2.0 eV/Å, indicating the dynamical stability of our system (for details, see Supplementary Fig. 2). In addition, we distort the crystal structures by creating a rotated Kagome lattice similar to ref. [56], and structural relaxation at different electric fields always removes the rotation distortion. Experimentally it has been found that both monolayer silicate and bilayer silica can be grown on various substrates with biaxial strains varying from $-5.6$ to

5.7%[44,48,52,53,64,65]. In addition, theoretical calculations have shown that defect-free 2D silica can be deformed up to a maximum strain of 10.4%, while for defective samples no abrupt material failure is observed at strains around 7.8–8.1%[63]. Regarding the electric field, by applying bias voltage between the tip of the scanning tunneling microscopy (STM) and the sample, a maximum electric field of 1.7 eV/Å can be obtained[66]. Therefore, both the strain and the electric field used in our calculations are experimentally feasible.

The most interesting regime is provided by tensile strain (exemplified with the 5% case in Fig. 1b). Under these conditions, the bandwidth of the three Kagome bands in group 1 decreases, reducing the frequency difference between the three bands at the Γ point. As a result, experimentally feasible electric fields can be used to drive phonon band inversion under strain. Similarly, the Kagome bands in groups 2 and 3 become closer at larger strains, again facilitating electric field manipulation.

Hereafter we focus on phonon dispersions at fixed strains under tunable electric field, because the strain is fixed by the material synthesis and determined by the substrate, while the electric field can be tuned using a gate voltage. We only focus on the three groups of Kagome bands because they are more sensitive to the electric field (see Supplementary Fig. 3 for the full phonon spectra of 5% and 7% strained monolayer silicates at different electric fields).

**Braiding in group 1.** We first study the braiding process of the Kagome bands in group 1 formed by the phonon branches $\mathcal{B}_{10,11,12}$ at 7% strained silicate. As shown in Fig. 2a, without an electric field, the Kagome bands at the Γ point are comprised of a single band $\mathcal{B}_{10}$ associated with a 1D irreducible representation (irrep) $\Gamma_1$ and a doubly degenerate band $\mathcal{B}_{11,12}$ associated with a 2D irrep $\Gamma_5$. Away from Γ, the doubly degenerate bands split, each carrying a different 1D irrep along the Γ-M and K-Γ high-symmetry lines.

Applying an electric field increases the frequency of the non-degenerate band $\mathcal{B}_{10}$ at Γ while reducing the frequency of the degenerate bands $\mathcal{B}_{11,12}$. The $\mathcal{B}_{10}$ vibrational mode at Γ corresponds to an out-of-plane displacement of silicon atoms and an opposite-direction displacement of oxygen atoms, and the $\mathcal{B}_{11,12}$ mode consists of both the in-plane displacements of silicon atoms and the out-of-plane displacements of oxygen atoms (for details, see the vibrational patterns in Supplementary Fig. 4). For the $\mathcal{B}_{10}$ mode, the two types of atoms carry out-of-plane Born effective charges of $+1.0$ and $-0.5$, respectively, and an out-of-plane electric field can enhance the opposite out-of-plane motions of two types of atoms with opposite signs of Born effective charge, which hardens the vibrational mode and increases the frequency of $\mathcal{B}_{10}$ at Γ. For the $\mathcal{B}_{11,12}$ mode at Γ, the out-of-plane electric field increases the out-of-plane distance between Si and O atoms, leading to weakened interatomic interactions and reduced phonon frequencies. Upon the electric field, the frequency of the lower $\mathcal{B}_{10}$ increases while the frequency of the higher $\mathcal{B}_{11,12}$ decreases, and a phonon band inversion takes place at an electric field of 0.7 eV/Å. As a result, new nodes are formed by the inverted bands with different irreps. For bands $\mathcal{B}_{10}$ and $\mathcal{B}_{11}$, a node forms near Γ along the K-Γ high-symmetry line as the two bands belong to different $\Lambda_1$ and $\Lambda_2$ irreps. On the other hand, there is no crossing point along Γ-M because the two bands belong to the same $\Sigma_1$ irrep. Due to the $C_6$ rotational symmetry of the system, there are six nodes in total within the gap $\Delta_{10}$ along the different K-Γ lines in the full Brillouin zone, as indicated by the yellow circles in Fig. 2b. For bands $\mathcal{B}_{11}$ and $\mathcal{B}_{12}$, six nodes form along Γ-M as they belong to the distinct irreps $\Sigma_1$ and $\Sigma_2$, while there is no crossing point along K-Γ since there the two

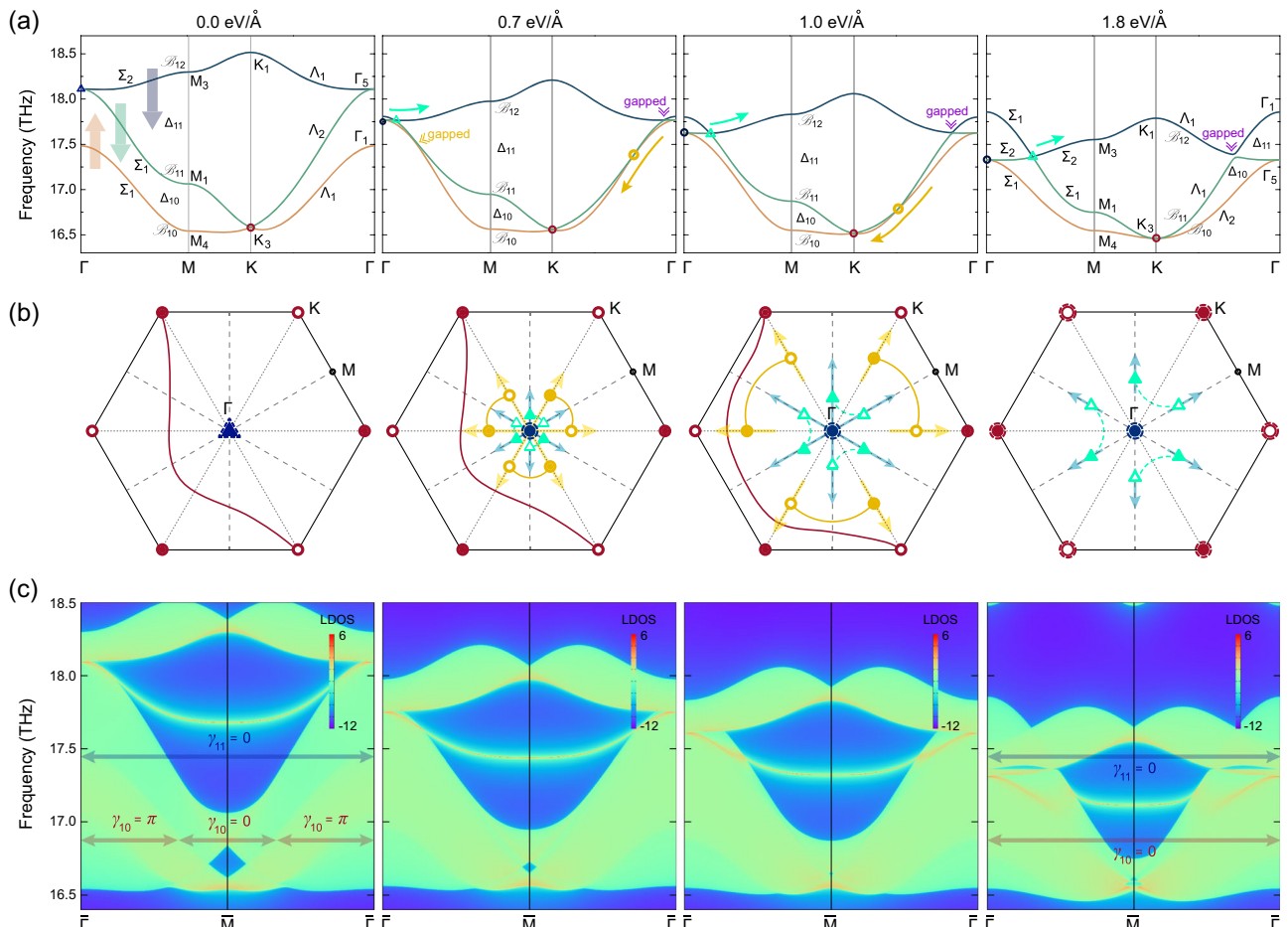

**Fig. 2 Braiding of phonons in group 1. a** Kagome bands and **b** their corresponding nodes in the 2D Brillouin zone formed by phonon branches $\mathcal{B}_{10-12}$ of 7% strained silicate under electric fields of 0.0 eV/Å, 0.7 eV/Å, 1.0 eV/Å, and 1.8 eV/Å. The topological configurations with nodes in different gaps (different symbols) and Dirac string configurations are shown in (**b**). The large blue triangle (at 0.0 eV/Å) and the large blue circles (at 0.7–1.8 eV/Å) with dashed boundary at Γ, as well as the large dark red circles with dashed boundary at K (at 1.8 eV/Å), are quadratic nodes. Note that crossing Dirac string in the same gap (same symbols) does not convert the charge and that moving nodes to the K point ensures that three charges meet in the extended zone. Hence moving the yellow nodes to K gives stable nodes with stable Euler class at K upon moving from the third to the last panel in (**b**). The corresponding phonon edge states on the (100) edge are shown in (**c**), with the $\{0, \pi\}$-quantised Zak phases $\{\gamma_n\}_{n=10,11}$ for gaps $\{\Delta_n\}_{n=10,11}$ indicated by the arrows.

bands belong to the same irrep $\Lambda_1$. As a result, six nodes are created near the Γ point within the gap $\Delta_{11}$ at 0.7 eV/Å, as indicated by the cyan triangles in Fig. 2b.

Further increasing the electric field enhances the phonon band inversion, such that the six nodes of gap $\Delta_{10}$ (yellow circles) move away from Γ towards the K points, and the six nodes of gap $\Delta_{11}$ (cyan triangles) move away from Γ towards the M points. At a threshold field of about 1.8 eV/Å, the $\Lambda_2$ band becomes lower than the other two $\Lambda_1$ bands over the whole K-Γ line, and the yellow nodes disappear upon reaching the K point. On the other hand, the cyan nodes remain in the middle of the Γ-M high-symmetry line even upon applying higher electric fields.

Throughout this entire process, there is also a nodal point in gap $\Delta_{10}$ at K with a 2D irrep $K_3$. This band crossing point remains nearly unchanged with varying electric field.

We now come to the topological characterisation of the above processes. In particular, we characterise the transfer of nodes from one gap to adjacent gaps with an associated transfer of patch Euler class and non-Abelian frame charges (see the 'Method' section for a detailed introduction of these topological concepts). Because of the *P6mm* space group, the system has $C_2$ rotation symmetry. In addition, in phonons the time reversal symmetry is automatically satisfied. Therefore, monolayer $Si_2O_3$ has $C_2T$

symmetry and its band nodes at neighbouring gaps can host non-Abelian charges. Although there might be some defects in monolayer silicate due to the imperfection of the growth processes[55,60,67], the impurities cannot destroy the non-Abelian charges as long as the $C_2T$ symmetry is preserved.

In the following, we label the frame charges and the patch Euler classes according to the gap to which the nodes they describe belong, i.e. for the $n^{th}$ gap with $n \in \{1, ..., 20\}$, we write the frame charges and the patch Euler classes as $\{\pm q_n\}$ and $\chi_n$, respectively.

The topological configurations can be determined by the numerical calculation of the patch Euler class for every single band crossing and for every pair of band crossings of the same gap. We then show how the original data of the computed patch Euler classes can be combined with the assignment of Dirac strings[11] to build the non-Abelian topological configuration of the nodes over the whole Brillouin zone (this is the *puzzle* approach detailed in 'Methods'). The results are then corroborated through the direct computation of the non-Abelian charge of nodes located in connected *multi-band subspaces*. The later requires the use of *partial-frames* (of the connected band subspaces) for which we present an algorithm in Methods.

Let us illustrate this strategy with the example of group 1 composed of three connected bands. Focusing on gap $\Delta_{10}$ in 7%

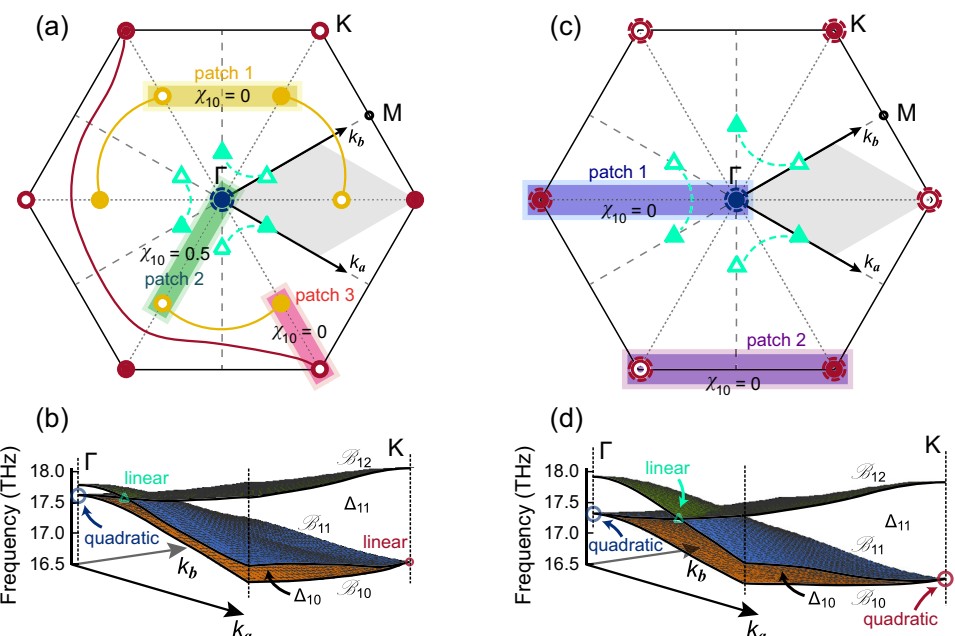

**Fig. 3 Euler class calculations. a** Patches in the Brillouin zone and their corresponding Euler class for 7% strained silicate under an electric field of 1.0 eV/Å. The Euler class $\chi_{10}$ is calculated from the phonon eigenvectors of $\mathcal{B}_{10}$ and $\mathcal{B}_{11}$ within the patches. The resultant patch Euler class in the yellow (patch 1), green (patch 2) and magenta (patch 3) areas are 0, 0.5 and 0, respectively. The large blue circle with dashed boundary at $\Gamma$ is a quadratic node with the patch Euler class of 1. **b** 3D band structures in the grey areas of (**a**), with two linear nodes —one (cyan triangle) in $\Delta_{11}$ along $\Gamma$-M and one (dark red circle) in $\Delta_{10}$ at K, as well as a quadratic node (blue circle) in $\Delta_{10}$ at $\Gamma$. These dispersions root in the stability of Euler class, meaning that a stable double node produces a quadratic dispersion. **c** Patches in the Brillouin zone to calculate the Euler class for 7% strained silicate under an electric field of 1.8 eV/Å. The patch Euler class in the blue (patch 1) and purple (patch 2) patches are both 0. The large blue circle with dashed boundary at $\Gamma$ and the large dark red circles with dashed boundary at K are quadratic nodes with the patch Euler class of 1. **d** 3D band structures in the grey areas of (**c**), with one linear node (cyan triangle) in $\Delta_{11}$ along $\Gamma$-M and two quadratic nodes in $\Delta_{10}$ at $\Gamma$ (blue circle) and K (dark red circle).

strained silicate under an electric field of 1.0 eV/Å, we compute the patch Euler class[11,13,68,69] from the numerically calculated phonon eigenvectors $|u_{10}\rangle$ and $|u_{11}\rangle$, of $\mathcal{B}_{10}$ and $\mathcal{B}_{11}$, respectively (see 'Methods'), for every single band crossing and for every pair of band crossings within the gap. In Fig. 3a, where we use the convention that the symbols of circle and triangle correspond to nodes in gap $\Delta_{10}$ and $\Delta_{11}$, respectively, we draw the patches that contain a pair of band crossings located at distinct regions of the Brillouin zone, with the calculated Euler classes indicated for all the patches. The Euler classes of the patches containing a single band crossing are indicated by the size of the symbols, i.e. the small circles indicate $|\chi_{10}| = 0.5$ (e.g. the nodes at K) and the large circles with dashed boundary indicate $|\chi_{10}| = 1$ (e.g. the node at $\Gamma$), and we use the convention that the fullness and openness of the symbols indicate the signs + and −, respectively. We note the correspondence between the dispersion around the nodes and their Euler class, i.e. the nodes with an Euler class of 0.5 exhibit a linear dispersion, while the nodes with an Euler class of ±1 appear to be quadratic (see also the section on the Euler class-dispersion relation below). We verify this by direct computation (see 'Methods' on the non-Abelian charges of partial-frames) showing that each Euler class of ±0.5 corresponds to a three-band partial-frame charge $q_{10} = \pm i \in \mathbb{Q}$, while the Euler class of ±1 corresponds to a three-band partial frame charge of $−1 \in \mathbb{Q}$.

The quadratic nodes can be viewed as two linear nodes merged together. This is explicitly verified under the breaking of $C_6$ where the double node at $\Gamma$ splits into two simple nodes (see the braid trajectory under broken $C_6$ symmetry in 'Methods'). Thus, for every band crossing and for every pair of band crossings within the gap, the patch Euler class takes value in the integers when it indicates an even number of stable nodes, or in the half-integers when it indicates an odd number of stable nodes within the patch,

as shown in Fig. 3a. Since the rotation of a patch by a symmetry of the point group $C_{6v}$ leads to an equal Euler class, we only need to consider the patches over the irreducible Brillouin zone. Importantly, the sign of the Euler class on each patch taken separately is gauge dependent, and similarly for the sign of the frame charges $q_{10}$ (see 'Methods'). This motivates the puzzle approach, which focuses on the *relative* stability of the nodes, because their relative stability is gauge invariant.

As a next step, we choose one initial patch containing a pair of band crossings, starting from patch 1 in Fig. 3a for which we obtain an Euler class of 0, and we assign a *signed* non-Abelian charge to each crossing. With an Euler class of 0, we can assign opposite charges to the pair of yellow nodes in patch 1 with no adjacent Dirac string crossing the patch. (Alternatively, we could assign the same charge to these two nodes and separate them by an adjacent Dirac string crossing the patch, see 'Methods' on the gauge freedoms.) The remaining yellow nodes, and the Dirac strings connecting them, are then fixed by the $C_6$ symmetry. For patch 2, an Euler class of $\chi_{10} = 0.5$ is compatible with the quadratic (blue) node of Euler class $\chi_{10}[\Gamma] = +1$ at $\Gamma$ and the linear (yellow) node of Euler class $\chi_{10}[\Gamma − K] = −0.5$ along $\Gamma$-K. For patch 3, $\chi_{10} = 0$, and we can assign opposite frame charges to the yellow node and the dark red node. Similarly, we then assign signed frame charges to all the other nodes, as well as Dirac strings that connect them. While there are relative gauge freedoms in doing so (see the 'Methods' section), once the charges of the initial patch is fixed, the charges for all neighbouring patches become fixed by consistency, like completing a puzzle. By repeating this process for all the patches, we get the complete topological configuration, as summarised in Fig. 3a. We remark that when a node in gap $\Delta_n$ crosses a Dirac string connecting nodes residing in *neighbouring gap* $\Delta_{n−1}$ or $\Delta_{n+1}$, the

sign of the charge in $\Delta_n$ changes. It should also be noticed that the configurations of the Dirac string is not unique, since moving the Dirac string flips the gauge sign of the eigenvectors locally. Nevertheless, the requirement of consistency in the assignment of the signed frame charges, together with the location of the Dirac strings, eventually guarantees the full indication of the relative stability of the nodes that is gauge independent.

A similar procedure can be applied to study the topological configurations for 7% strained silicate under an electric field of 1.8 eV/Å. As shown in Fig. 3c, patch 1 in $\Delta_{10}$ contains a dark red node at K, a quadratic blue node at $\Gamma$, and a Dirac string connecting the cyan nodes in $\Delta_{11}$. With an Euler class $\chi_{10} = 0$, the dark red node at K must carry the same frame charge with the blue node at $\Gamma$. In addition, patch 2 has an Euler class of 0, indicating that the neighbouring dark red nodes carry opposite frame charges. For the cyan nodes in $\Delta_{11}$, there is no nearby Dirac string in $\Delta_{10}$ during the whole braiding process, and their frame charges remain the same.

Once an initial topological configuration is known, we can determine any future topological configurations resulting from a band inversion by simply applying the rules of braiding (see 'Methods') together with the constraints of the crystal symmetries. Figure 2b represents the conversion of the topological configuration through the band inversion within the bands of group 1. We start with the topological configuration at 0.0 eV/Å showing two linear nodes in $\Delta_{10}$ at K ($-q_{10}$, open dark red circle) and K' ($q_{10}$, closed dark red circle) connected by a Dirac string (dark red line), and one quadratic node in $\Delta_{11}$ at $\Gamma$ (large blue triangle with dashed boundary) with a patch Euler class $\chi_{11}[\Gamma] = +1$, corresponding to a three-band frame charge $q[\Gamma] = -1$ (computed for a base loop encircling the $\Gamma$ point while avoiding the nodes at the K points).

From the irreps of the bands given in Fig. 2a we have predicted the formation of symmetry protected nodes on the $\Gamma$-M lines in $\Delta_{11}$ and on the $\Gamma$-K lines in $\Delta_{10}$ under the inversion of the bands at $\Gamma$. Figure 2b shows the complementary topological configurations of the nodes. It is important to note the relative signs of the charges, which are not accidental. Indeed, let us imagine the reverse band inversion process, i.e. from 1.0 eV/Å to 0.0 eV/Å in Fig. 2b while relaxing the $C_6$ symmetry but conserving $C_2T$ symmetry, i.e. allowing the nodes to move freely on the $C_2T$ symmetric plane where they are pinned. We can first recombine the circles in $\Delta_{10}$ by bringing the yellow nodes of the $\Gamma$–K lines to the $\Gamma$ point. By doing so, the three filled yellow circles must cross the dashed-line (cyan) Dirac strings and the three open triangles are crossed by the full-line (yellow) Dirac strings, which implies a flip of their frame charges (see the Method section). We thus get six closed cyan triangles inside the Brillouin zone, together with six open yellow circles and two filled blue circles (obtained after splitting the quadratic node at $\Gamma$). The circle nodes recombine, leaving four open circles and six closed triangles. Braiding two of the open circles with two of the closed triangles, we get two pairs of circles and triangles that can be annihilated (see the Method section), leaving a single pair of closed triangles, i.e. we are back at the topological configuration at 0.0 eV/Å in Fig. 2b.

In 'Methods', we compute the trajectory of the nodes through the band inversion when the $C_6$ symmetry is broken, using an effective three-band tight-binding model. This very explicitly reveals the braiding process involved in the transfer of nodes from one gap to the other. When $C_6$ symmetry is recovered, the braid trajectories are collapsed onto the high-symmetry points (here $\Gamma$), leading to the necessary formation of triply-degenerate points during the inter-gap transfer of nodes.

By moving the nodes in $\Delta_{10}$ (yellow circles) on the $\Gamma$–K lines to the K points, we obtain the topological configuration of the fourth

panel in Fig. 2b with quadratic nodes at K ($\chi_{10}[K] = +1$) and K' ($\chi_{10}[K'] = -1$).

Summarising the general philosophy behind the determination of topological phase transitions in the non-Abelian topological phases protected by $C_2T$ symmetry: we first have a puzzle "game" followed by a braiding "game".

**The Euler class-dispersion relation.** We also calculate the 3D band structures for 7% strained silicate under an electric field of 1.0 eV/Å in Fig. 3b to demonstrate the quadratic node in $\Delta_{10}$ at $\Gamma$, as well as the linear nodes in $\Delta_{10}$ at K and in $\Delta_{11}$ along $\Gamma$–M, which relates to their frame charges. We note in Fig. 3b the quadratic dispersion in all directions of the double degeneracy at $\Gamma$.

Figure 3d shows the conversion of the dispersion of the nodes at K and K' from linear to quadratic when increasing the electric field from 1.0 eV/Å to 1.8 eV/Å. Below, we elaborate further on the relation between the patch Euler class of a single band crossing and the power-law (linear or quadratic) of the dispersion at the crossing.

We make the general observation that for any single band crossing at a momentum $\mathbf{k}_0$, the patch Euler class, $\chi[\mathbf{k}_0]$, determines the lower bound of the degree of the dispersion of the Bloch eigenvalues, $\lambda(\mathbf{k})$, at the band crossing: defining the order of the leading term in a Taylor expansion of the Bloch eigenvalues at $\mathbf{k}_0$ as $\alpha$, we get $2\chi[\mathbf{k}_0] \leq \alpha$[15]. For instance, a single node with Euler class 0.5 must exhibit a dispersion at least linear ($2\chi = 1 \leq \alpha$). We call this the Euler class-dispersion relation. In that regard it is crucial to remember that the eigenvalues of the dynamical matrix are frequencies squared ($\lambda = E^2$) while the phonon band structures are plotted as a function of frequency ($E = \sqrt{\lambda}$). Defining by $\beta$ the order of the leading term of a Taylor expansion of the phonon frequencies at a band crossing, the Euler class-dispersion relation thus gives $\chi[\mathbf{k}_0] \leq \alpha/2 = \beta$. Very interestingly, except for the lowest phonon bands at $\Gamma$[26], the dispersion of all the other band crossings always exhibits an order strictly higher than their Euler class lower bound, i.e. we find $2\chi[\mathbf{k}_0] \leq \beta$. This implies that the hoping terms of the Wannierised dynamical matrix are dominated by long-range hoping processes, contrary to electronic systems where the short-range hoping processes are strongly dominant due to screening.

**Braiding in groups 2 and 3.** We next focus on the band inversion and the resulting braiding of the phonon bands in groups 2 and 3 under 5% strain, which become connected as the electric field increases. As shown in Fig. 4a, at a negative electric field of $-0.3$ eV/Å, the two groups of phonon bands are well-separated. At $-0.2$ eV/Å, the highest band in group 2 ($\mathcal{B}_{16}$) and the lowest band in group 3 ($\mathcal{B}_{17}$) touch at K, and six nodes (cyan circles in Fig. 4b) are created along K–$\Gamma$ as the two bands belong to the different irreps $\Lambda_1$ and $\Lambda_2$ (there are also six nodes created on the M–K line at 0.0 eV/Å, which annihilate in pairs at the M point upon increasing the electric field, a process that we do not show here). The cyan nodes move towards $\Gamma$ upon increased electric field, and touch the $\Gamma$ point around 0.5 eV/Å causing a band inversion at $\Gamma$. Instead of being annihilated, these six nodes bounce back along their original path.

In addition to the cyan nodes formed by $\mathcal{B}_{16}$ and $\mathcal{B}_{17}$, there are also six nodes (purple squares) formed by $\mathcal{B}_{17}$ and $\mathcal{B}_{18}$, as well as six nodes (yellow triangles) formed by $\mathcal{B}_{15}$ and $\mathcal{B}_{16}$, along the $\Gamma$–M high-symmetry line, see Fig. 4b. They are all created at about 0.5 eV/Å near the $\Gamma$ point, and move towards M with increasing electric field. The yellow nodes move much faster than the purple ones, which is due to stronger band inversion.

We now discuss the topological features of the band inversion between the bands in groups 2 and 3. Again, the determination of

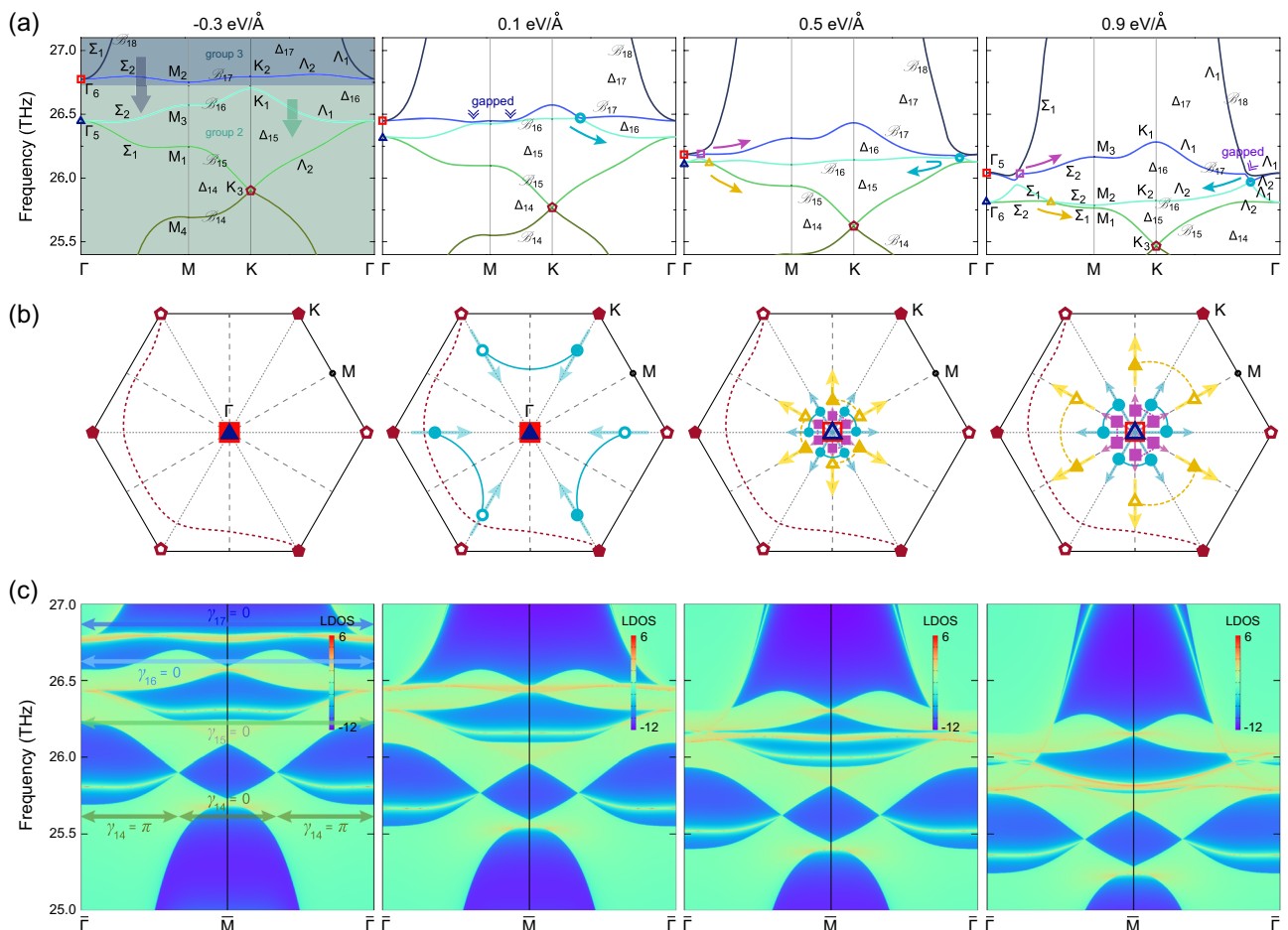

**Fig. 4 Braiding of phonons in groups 2 and 3. a** Phonon branches $\mathcal{B}_{14-18}$ and their corresponding nodes of 5% strained silicate under electric fields of $-0.3$ eV/Å, 0.1 eV/Å, 0.5 eV/Å, and 0.9 eV/Å. The panels in **b** show the nodes and their topological charges. As before, different symbols characterise topological charges in different gaps, whereas open versus closed symbols indicate the sign of the charge. Dirac strings are drawn as lines. The large blue triangles and the large red squares at $\Gamma$ are quadratic nodes. The corresponding phonon edge states on the (100) edge are shown in (**c**), with the $\{0, \pi\}$-quantised Zak phases $\{\gamma_n\}_{n=14, \ldots, 17}$ for the gaps $\{\Delta_n\}_{n=14, \ldots, 17}$ indicated by the arrows.

the initial topological configuration (i.e. fixing the frame charges and the Dirac strings) is achieved through the puzzle construction detailed in the previous section. The transitions to the other topological configurations upon the band inversions then follow the rules of (symmetry-constrained) braiding. First, $\mathcal{B}_{16}$ and $\mathcal{B}_{17}$ are inverted at the K point. The irreps of Fig. 4a predict the formation of six nodes along the K–$\Gamma$ lines and six nodes along the M–K lines (at 0.0 eV/Å, not shown here), all in $\Delta_{16}$ and protected by symmetry. Figure 4b shows the charges for the nodes in $\Delta_{16}$ (cyan circles), and for the adjacent quadratic nodes at $\Gamma$ in $\Delta_{15}$ ($\chi_{15}[\Gamma] = +1$, large blue filled triangle) and in $\Delta_{17}$ ($\chi_{17}[\Gamma] = +1$, large red filled square). Let us imagine the reverse band inversion process from 0.1 eV/Å to $-0.3$ eV/Å, while we relax the $C_6$ crystalline symmetry but conserve the $C_2T$ symmetry (i.e. allowing the nodes to move freely on the $C_2T$ symmetric plane). After bringing all the circles to K and K′ they must annihilate, leaving a pair of closed Dirac strings (loops) behind. Since one Dirac string marks a flip of gauge sign of the eigenvectors (see the 'Method' section), the merging of two Dirac strings corresponds to the identity, such that the two closed Dirac strings can merge and annihilate.

The conversion from the first to second panel in Fig. 4b follows the creation and the moving of the nodes on the K–$\Gamma$ line from K towards $\Gamma$. When the nodes in $\Delta_{16}$ (cyan circles) reach $\Gamma$, the band inversion happens between the two doubly-degenerate irreps $\Gamma_5$

and $\Gamma_6$. From the irreps of the bands (Fig. 4a) we predict the formation of six nodes (yellow triangles) on the $\Gamma$–M lines in $\Delta_{15}$, six nodes (cyan circles) on the K–$\Gamma$ lines in $\Delta_{16}$, and six nodes (purple squares) on the $\Gamma$–M lines in $\Delta_{17}$. The corresponding topological configurations are shown in the third panel of Fig. 4b, from which we can verify that the reversing of the band inversion by recombining the nodes must lead to the second panel. The linear nodes at the K point (full and open dark red pentagons) together with their Dirac string (dashed dark red line) in $\Delta_{14}$ affect the behaviour of the triangle nodes when they reach the M point upon applying higher electric fields, i.e. these cannot annihilate and instead scatter apart from one another (not shown here).

Similar to the group 1, the non-Abelian frame charges within the groups 2 and 3, corresponding to the patch Euler class of the single band crossings, can be verified through a direct computation using the algorithm presented in 'Methods' which adapts the results of ref. [10] to partial frames. We note in passing that while the band inversion in group 1 at $\Gamma$ is mediated by a triple-degenerate point with a frame charge of $q = -1$[15,70,71], the band inversion at $\Gamma$ between group 2 and 3 is mediated by a quadruple-degenerate point with a total frame charge $q = (-1) * (-1) = +1$. Even though the frame charge of the quadruple-degenerate node is trivial, because of the crystalline symmetries it must be formed by the superposition of two quadratic nodes, each with a nonzero

patch Euler class $|\chi_{15}| = |\chi_{17}| = 1$, and six linear nodes with the frame charges $\pm q_{16}$.

**Edge states.** We also study the evolution of topological edge states with varying electric field by calculating the surface local density of states (LDOS) from the imaginary part of the surface Green's function[72]. Figure 2c shows the edge states of 7% strained silicate. On the (100) edge (i.e. the zigzag direction for the Si atoms), there is an edge arc connecting two adjacent nodes at the neighbouring $\overline{\Gamma}$ points. Upon increasing electric field, new band crossing points appear around $\overline{\Gamma}$. At 1.8 eV/Å, there are two clear projections of the new Dirac cones on the (100) edge around 17.36 THz. We calculate the Zak phase $\gamma$ (i.e. the Berry phase along a non-contractible path of the Brillouin zone which is here quantised to $\{0, \pi\}$ by the $C_2T$ symmetry) in gap $\Delta_{10}$ and $\Delta_{11}$ at 0.0 and 1.8 eV/Å. As shown in Fig. 2c, a Zak phase of zero corresponds to the emergence of the edge states, while $\gamma = \pi$ leads to vanishing edge states. This indicates that the band Wannier states (the Wannier states are the Fourier transform of the Bloch eigenstates) have their center (the "band centers") shifted from the center of the atomic Wannier states (the "atomic centers"), leading to a charge anomaly and, following, the localisation of an edge state. Furthermore, the fact that the edge states appear when the Zak phase is zero simply means that the band centers occupy the center of the unit cell, while the atomic centers lie on the boundary of the unit cell[15,73]. Indeed, the $\mathbb{Z}_2$ quantised Berry phase is a good quantum number in 1D, which indicates the band center (i.e. its Wyckoff position)[74].

We can actually predict the Zak phase for the given edge termination from the the bulk topology. The Zak phase at a fixed gap is given (modulo $2\pi$) by the parity of the number of Dirac strings that are crossed by the straight path of integration that is perpendicular to the edge axis, i.e. for the (100) edge these are the paths $l_{k_\parallel} \in \{k_\perp(\mathbf{b}_1 - \mathbf{b}_2) + k_\parallel(\mathbf{b}_1/2 + \mathbf{b}_2/2) | k_\perp \in [0, 1]\}$ at a fixed $k_\parallel \in [0, 1]$, with $k_\parallel$ the coordinate of the horizontal axis in Fig. 2c and Fig. 4c[15].

Figure 4c shows the topological edge states of 5% strained silicate on the (100) edge. Besides the edge arc that connects two adjacent nodes at the neighbouring $\overline{\Gamma}$ points around 26.45 THz, there is also a new edge arc connecting two adjacent nodes around 25.90 THz located at the projections of the neighbouring K points with 2D irrep $K_3$. Under an electric field of 0.9 eV/Å, extra edge arcs emerge near $\overline{\Gamma}$ at 25.97 THz. The arc connecting $K_3$ moves downwards upon increasing electric field and is robust. For the four-band braiding processes, the Zak phase argument fails, as the Zak phases $\gamma_{14-17}$ in Fig. 4c do not show a consistent behaviour. We stress that, apart from effective Zak phase diagnoses[15,75], the full multi-gap bulk-boundary correspondence remains an open question. Hence, we take here a "spectator" view by directly visualising the edge states but without addressing the fundamental mechanisms behind them.

**Experimental signature: Raman spectra.** We finally propose that the band inversion processes described above can be directly observed experimentally by following the evolution of the Raman spectrum of the material. All the relevant modes are Raman active (for details, see Supplementary Fig. 5). We calculate the evolution of the Raman spectrum associated with the modes involved in the two braiding processes described in Figs. 2 and 4. Figure 5a shows the two Raman modes of the three Kagome bands in group 1. Without an electric field, $\mathcal{B}_{10}$ at $\Gamma$, with 1D irrep $\Gamma_1$, belongs to the Raman active $A_1$ mode at 583.1 cm$^{-1}$. $\mathcal{B}_{11}$ and $\mathcal{B}_{12}$ at $\Gamma$, with 2D irrep $\Gamma_5$, correspond to the $E_2$ mode, and are also Raman active at 604.1 cm$^{-1}$. The Raman peak of the $E_2$ mode is stronger than that of the $A_1$ mode. With increasing electric field, the frequency of the stronger $E_2$ mode decreases, until reaching the critical field of

0.7 eV/Å, where its phonon frequency of 592.7 cm$^{-1}$ becomes lower than that of the weaker $A_1$ mode (594.0 cm$^{-1}$). Further increasing the electric field enlarges the frequency difference between the two Raman active modes.

For the two Raman modes involved in the braiding processes between groups 2 and 3, the band inversion between the Raman modes with different intensities is also clearly visible. As shown in Fig. 5b, both the $E_2$ and $E_1$ modes, corresponding to the 2D irreps of $\Gamma_5$ and $\Gamma_6$, respectively, are Raman active, and the $E_1$ peak with a higher frequency of 893.2 cm$^{-1}$ has a stronger intensity. Although the frequencies of both the stronger $E_1$ mode and the weaker $E_2$ mode become lower as the electric field increases, the frequency of the $E_1$ mode decreases much faster than that of the $E_2$ mode. As a result, the two bands invert at 0.5 eV/Å.

These calculations show that Raman spectroscopy can be a promising tool for characterising the band inversion processes and the accompanying non-Abelian braiding of phonons in 2D materials such as monolayer silicate. Alternatively, inelastic neutron scattering[36,76,77] and inelastic X-ray scattering[33] can be used to observe the bulk band crossings directly, while high resolution electron energy loss spectroscopy[78] can be used to observe the topological surface states.

In conclusion, we show that the phonon bands in layered silicates provide a versatile platform to observe new multi-gap topologies. We find that under experimentally feasible strain and/ or external electric field conditions, the phonon bands can exhibit the required braiding processes to induce multi-gap topological phases characterised by non-Abelian frame charges and Euler class. Given the feasibility of the proposed material and experimental setup, we hope that this study provides an impetus to investigate phonon bands, and this material realisation in particular, to experimentally observe multi-gap topology.

## Methods

**Computational details.** First principles calculations using density functional theory are performed with the Vienna ab initio Simulation Package (VASP)[79,80]. The generalised gradient approximation (GGA) with the Perdew-Burke-Ernzerhof (PBE) parameterisation is used as the exchange-correlation functional[81]. A plane-wave basis with a kinetic energy cutoff of 800 eV is employed, together with a $9 \times 9 \times 1$ **k**-mesh to sample the electronic Brillouin zone. The self-consistent field calculations are stopped when the energy difference between successive steps is below $10^{-8}$ eV, and the structural relaxation is stopped when forces are below $10^{-3}$ eV/Å. A vacuum spacing larger than 20 Å is used to eliminate interactions between adjacent layers.

The ionic positions with and without electric fields are fully relaxed while the lattice constants are fixed, corresponding to material synthesis on substrates at fixed strain. When applying an external electrostatic field, dipole corrections are included to avoid interactions between the periodically repeated images.

For the phonon calculations, the force constants are evaluated using the finite differences method in a $3 \times 3 \times 1$ supercell with a $3 \times 3 \times 1$ electronic **k**-mesh using VASP. The phonon dispersion is then obtained using PHONOPY[82]. Convergence tests have been performed comparing supercells of sizes between $3 \times 3 \times 1$ and $6 \times 6 \times 1$ (for details, see Supplementary Fig. 6). In 2D monolayers, no splitting between the longitudinal and transverse optical phonons (LO-TO splitting) occurs at $\Gamma$, and only the slope of phonon bands changes[83]. Therefore, we ignore LO-TO splitting as it does not influence the phonon band crossings.

The Euler class of nodes is calculated by employing the Wilson-loop method to calculate the Wannier charge center flow, i.e., the monopole charge of a node[17,84,85]. The patch Euler class is obtained by integrating phonon eigenvectors within a patch around two nodes, and a unitary rotation is applied to make the eigenvectors real. The Euler class is computed from a pair of bands on a rectangular region that contains the nodes using the code in ref. [86], as described in ref. [13]. We then assign the frame charges to all the nodes, as well as Dirac strings that connect the frame charges. We repeat this process for all the patches and get the complete topological configuration. As explained in the following subsection, the concepts of non-Abelian frame charges, patch Euler class, and Dirac strings enable a descriptive and predictive representation of the topology.

The phonon edge states are obtained using surface Green's functions as implemented in WANNIERTOOLS[72].

**Topological invariants and topological configurations.** Combining topology with space group symmetries has resulted in a myriad of single-gap topological

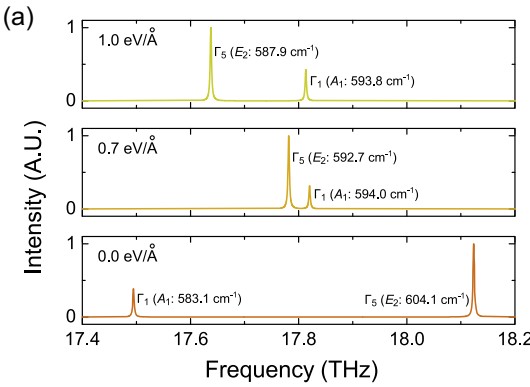
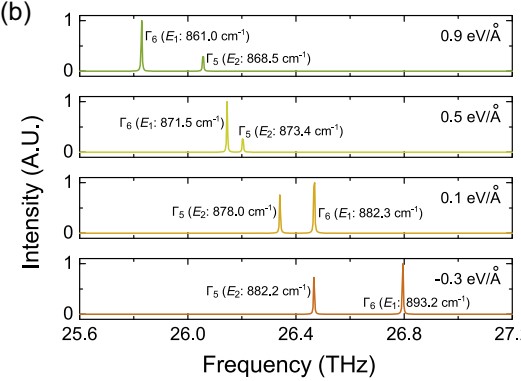

**Fig. 5 Raman signature.** Simulated Raman spectra (A.U. = arbitrary units) for **a** 7% and **b** 5% strained monolayer silicate Si$_2$O$_3$ under various electric fields.

characterisations of materials[4–9,19,87–105]. In particular, a rather universal framework has been established to study single-gap topology: the determination of topologically inequivalent band structures using the band representations at high-symmetry points in the Brillouin zone[4–6], an approach that matches the full K-theory result in some scenarios[4,7]. These momentum space constraints can be compared to real space constraints, resulting in versatile classification schemes to characterise topological materials by identifying which of these combinations have an atomic limit[8,9]. These approaches have been supplemented with an alternative real space decomposition of topological states, giving rise to topological crystals[106–109].

In this section we move beyond the single-gap limit to multi-gap topologies instead, and describe the key concepts used to determine the topological configurations reported in the main text.

*Non-Abelian frame charges*. We start with a brief review of the non-Abelian frame charges[10]. At the end of the 'Methods' section, we then present a simple algorithm to compute the non-Abelian charges of an $M$-band subspace within a $N$-band system with $M \leq N$.

Let us first consider a three-band system where the three bands are non-degenerate almost everywhere. Labelling the eigenfrequencies $E_1 \leq E_2 \leq E_3$, we call $\Delta_1$ the partial frequency gap between the bands 1 and 2 ($E_1 \leq E_2$) and $\Delta_2$ the partial frequency gap between the bands 2 and 3 ($E_2 \leq E_3$). Because of the presence of $C_2T$ symmetry with $[C_2T]^2 = +1$, the Bloch Hamiltonian for the three-band system can be rotated into a basis in which it is real and symmetric[13], such that the corresponding Bloch eigenvectors $\{u_n\}_{n=1,2,3}$ are real and form an orthogonal matrix $R = (u_1u_2u_3) \in O(3)$, i.e. once oriented a *frame*[110]. Because the eigenvectors are real, the phase freedom of complex eigenvectors turns into a $\pm$ sign freedom. As a result, the classifying space of the three-band frames of eigenvectors is given by the real flag manifold $O(3)/O(1)^3 \cong SO(3)/D_2$[10]. Nodal points are characterised by the elements of $\pi_1[SO(3)/D_2] = \mathbb{Q} = \{+1, \pm i, \pm j, \pm k, -1\}$[10], i.e. the quaternion group. The quaternion group is non-Abelian since the elements $\{i, j, k\}$ anti-commute. We can then assign the non-Abelian quaternion charges to the nodes[10], e.g. we can take $\pm i$ for each linear node of gap $\Delta_1$, $\pm j$ for each linear node of gap $\Delta_2$, $\pm k$ for two linear nodes with one in each gap, and $-1 = i^2 = j^2 = k^2$ for two nodes with equal charge within the same gap[10]. Importantly, conjugation of a non-Abelian charge $q \in \{i, j, k\}$ by another captures the effect of braiding the corresponding node around the node of an adjacent gap, e.g. taking the node of charge $q = i$ in gap $\Delta_1$, its charge becomes $(-j)*(+i)*(+j) = -i$ after braiding it around the node of charge $j$ in gap $\Delta_2$[10].

It is important to note that, as elements of the first homotopy group, the non-Abelian charges of band nodes acquire an unambiguous meaning only once a fixed base point and an oriented base loop are chosen (as implied by the definition of the first homotopy *group*). Furthermore, the sign of the charges $q \in \{i, j, k\}$ also depends on the choice of a gauge at the base point of the loop[10,13,111]. We come back to the question of the gauge freedom when we introduce the Dirac strings below.

While the charge $-1$ can be readily characterised in terms of the frame $R \in SO(3)$ itself, i.e. as the generator of $\pi_1[SO(3)] = \mathbb{Z}_2$[112], it represents all the non-contractible loops within $SO(3) \cong \mathbb{D}^3/\sim$ (where $\mathbb{D}^3$ is the 3D unit ball and $\sim$ is the equivalence relation for antipodal points), the non-Abelian elements on the contrary, are obtained through the lifting of the frame to the spin group $Spin(3) = SU(2)$[10], i.e. the double (universal) covering space of the orthogonal group (see below).

Generalising to an arbitrary number of bands, the classifying space of real symmetric Hamiltonians with all the eigenfrequencies gapped is given by the complete Flag manifold $O(N)/O(1)^N \cong SO(N)/S[O(1)^N]$[10,111]. The non-Abelian charges of a frame with rank $N$ over a loop are then given by the first homotopy group $\pi_1[SO(N)/S[O(1)^N]] = \pi_1[Spin(N)/\overline{P}_N] = \overline{P}_N$, where $\overline{P}_N$ is isomorphic to the Salingaros' vee group[10,14]. $P_N = S[O(1)^N]$ is the group of all gauge transformations of $N$ ordered orthonormal eigenvectors that preserve the handedness of the frame there are forming together. More precisely, $P_N$ is the discrete group generated by the $\pi$ rotation of each pair of eigenvectors, i.e. $(u_1, \ldots, u_{n_1}, \ldots, u_{n_2}, \ldots, u_N) \rightarrow (u_1, \ldots, -u_{n_1}, \ldots, -u_{n_2}, \ldots, u_N)$ for all pairs $(n_1, n_2)$ with $1 \leq n_1 < n_2 \leq N$[10]. $\overline{P}_N < Spin(N)$ is then obtained as the double group of $P_N < SO(N)$[10]. While $\overline{P}_N$ are rather complicated objects, as far as we are concerned it is enough to know that for each gap $\Delta_n$, we can assign a pair of conjugated elements $\{q_n, -q_n\} \in \overline{P}_N$ to every simple node within $\Delta_n$, and that these, together with the adjacent charges, i.e. $\{\pm q_{n-1}, \pm q_n, \pm q_{n+1}\}$, mimic the behaviour of the quaternion frame charges $\{\pm i, \pm j, \pm k\}$. Indeed, two nodes coming from gaps that are not adjacent do not interact, as reflected by the commutation $q_nq_m = q_mq_n$ whenever $|n - m| \geq 2$[10,14]. Again, the charge $-1$ characterises the presence of two nodes with the same non-Abelian charge within one gap, while $+1$ indicates that there is no stable nodes.

Importantly, the charge of a node for distinct trajectories of the base loop, as well as the charge of multiple nodes, are obtained by following the rules of composition of paths and the corresponding multiplication rule of the non-Abelian frame charges[10]. While this approach becomes rather cumbersome when many nodes must be considered at the same time, the 1D topology (i.e. coming from the first homotopy group) can be refined by a quasi-2D invariant which is also at the basis of a more convenient method for the determination of the topological configurations of any nodal phase. Namely, we use below one approach[15] based on the *patch Euler class*[11,13,86], and the *Dirac strings*[11] connecting all the nodes of the same gap in pairs. As we discuss below, the Dirac strings are also a way to make the gauge choices explicit, while these are not physical observables.

*Patch Euler class*. As the *real* equivalent to the Chern class for complex Hamiltonians, the Euler class classifies the 2D topology of real Hamiltonians. Different from the frame charges that apply to at least three bands, the Euler class applies only to two-band subspaces, i.e. given two adjacent bands, say $\mathcal{B}_n$ and $\mathcal{B}_{n+1}$, that are isolated from the other bands by frequency gaps above and below, that is with $E_{n-1}(\mathbf{k}) < E_n(\mathbf{k}) \leq E_{n+1}(\mathbf{k}) < E_{n+2}(\mathbf{k})$ for all $\mathbf{k} \in BZ$. With real eigenvectors, the two-band Berry connection take values in the orthogonal Lie algebra $SO(2)$, i.e. these are $2 \times 2$ skew-symmetric matrices. We then define the Euler 2-form $Eu = d\mathbf{a} = dPf\mathcal{A}$ with $\mathcal{A}_{ab} = \langle u_a, \mathbf{k}|du_b, \mathbf{k}\rangle = \mathbf{A}_{ab} \cdot d\mathbf{k} = \sum_{i=1,2}\langle u_a, \mathbf{k}|\partial_{k_i}u_b, \mathbf{k}\rangle dk_i$ where we have set $A_{ab}^i = \langle u_a, \mathbf{k}|\partial_{k_i}u_b, \mathbf{k}\rangle$ ($A^i \in SO(2)$). This gives $Eu = (\langle\partial_{k_1}u_a, \mathbf{k}|\partial_{k_2}u_b, \mathbf{k}\rangle - \langle\partial_{k_2}u_a, \mathbf{k}|\partial_{k_1}u_b, \mathbf{k}\rangle)dk_1 \wedge dk_2$, where $a, b$ are the band indices which we take as $n, n+1$ below[11,13,68,69]. The Euler class is then given through the integration over the 2D Brillouin zone $\chi_n \equiv \chi[\{\mathcal{B}_n, \mathcal{B}_{n+1}\}] = (1/2\pi)\int_{BZ}Eu \in \mathbb{Z}$. The even integer $2|\chi_n| \in 2\mathbb{Z}$ gives the number of stable nodal points formed by the two bands $\mathcal{B}_n$ and $\mathcal{B}_{n+1}$[11,13,68,111]. These nodes can only be annihilated upon a braiding operation requiring a band inversion with at least a third band[11,113].

While the Euler class introduced above requires that the two bands under consideration should be isolated by a gap from the other bands over the whole Brillouin zone, very conveniently we can define a *patch* Euler class[11,13] that gives the number of stable nodes between two bands over one patch of the Brillouin zone $\mathcal{D} \subset BZ$, i.e. $\chi_n[\mathcal{D}] \equiv \chi[\{\mathcal{B}_n, \mathcal{B}_{n+1}\}; \mathcal{D}] = (1/2\pi)[\int_{\mathcal{D}}Eu - \oint_{\partial\mathcal{D}}\mathbf{a}] \in \mathbb{Z}$ where the Euler connection 1-form $\mathbf{a} = Pf\mathcal{A} \cdot d\mathbf{k}$ is integrated over the contour of the patch $\partial\mathcal{D}$. This number can be computed with an algorithm available in a MATHEMATICA NOTEBOOK at ref. [13,86].

Importantly, the patch Euler class is not independent of the frame charges. Considering the two adjacent bands $\{\mathcal{B}_n, \mathcal{B}_{n+1}\}$, the presence of a single linear node within a patch will give rise to a frame charge $(\pm)q_n$ and a patch Euler class of $\chi_n = (\pm)1/2$. In the case of two nodes of equal charge, we get $[(\pm)q_n]^2 = q_n^2 = -1$ and $\chi_n = \pm 1$. More generally, integer (half-integer) patch Euler classes indicate an

even (odd) number of nodes with the same frame charge through the equivalence

$$\pm 2|\chi_n| = N_{\pm q_n} \in \mathbb{N}, \qquad (1)$$

where $N_{\pm q_n}$ is the number of nodes with frame charge $\pm q_n$ in gap $\Delta_n$. We thus conclude that, in a two-band subspace, the patch Euler class greatly refines the frame charges since it captures the indefinite accumulation of nodes with equal charge between the two bands, compared to the $\{1, q_n, -q_n, -1\}$ frame charge classification of the nodes in a fixed gap $\Delta_n$.

We note that the Euler class cannot be computed when a band crossing point involves more than two degenerate bands (i.e. an $M$-fold crossing with $M > 2$). In such a case the direct computation of the frame charge cannot be avoided and it is complementary to the Euler class. Nevertheless, such threefold or fourfold band crossings are accidental in silicate, i.e. these are realised at a special value of the parameters that control the band inversions. Therefore, by choosing an appropriate initial phase with no higher degeneracies, all the frame charges of the nodes can be deduced from the computation of patch Euler classes only. We have confirmed this through the direct computation of the non-Abelian frame charges from partial frames, see the last section of 'Methods'.

One crucial observation is that the sign of the Euler class within each patch is gauge dependent. Indeed, flipping the gauge sign of one of the two eigenvectors used to compute a patch Euler class does flip the total sign of the patch Euler class. (Similarly, the frame charges depend on an initial choice of gauge at the base point of the loop, see the last section of Methods.) On the contrary, the absolute value of the Euler class captures the relative stability of the nodes within the patch, and is thus gauge invariant.

Our strategy bellow is to determine the global topological configuration of an initial nodal phase through the computation of Euler patch only (assuming that the phase does not contain accidental $N$-fold band degeneracies with $N > 2$). For this we will first systematically compute the patch Euler classes of all single band crossings. We do this effectively by defining a small neighborhood around each node such that it avoids all the other nodes. Then, considering each gap separately, we compute the patch Euler class for all possible pairs of band crossings within the gap.

Before we proceed, we need to overcome the gauge dependence of each patch Euler class (frame charge) taken separately. This is done by fixing a global gauge which necessitates the introduction of *Dirac strings*[11].

*Dirac strings.* Lying at the core of the computation of the patch Euler class is the necessity to regularise the gauge of the eigenvectors[10,13] in order to make them smooth almost everywhere within the patch (which is required for the definition of the Euler differential form). (This is also true for the computation of the frame charges, which requires a parallel transport over the base loop, see the last section of Methods.) The presence of nodes induces an obstruction to obtain fully smooth eigenvectors though. Indeed, whenever two simple nodes (i.e. each with an absolute Euler class of 1/2) are created within one gap, in which case the patch Euler class of the pair of nodes is 0, the $\pi$ Berry phase carried by each node indicates the presence of a $\pi$-disclination line connecting the two nodes[11], i.e. the eigenvectors must jump by a gauge sign flip over this line. This is represented by a Dirac string in the plane connecting the two nodes within the same gap[11], in analogy with the Dirac string connecting Weyl points in 3D in the complex case. Similarly, if we consider a patch containing a pair of simple nodes of equal frame charge, i.e. the patch Euler class is $\pm 1$, each node carries a $\pi$ Berry phase still and a Dirac string must again connect them. On the contrary, a single band crossing with a patch Euler class of $\pm 1$, i.e. a double node like those found at $\Gamma$, carries a 0 Berry phase (Berry phases are only defined modulo $2\pi$) and it is not connected to any Dirac string.

This assignment of Dirac strings within each patch can be readily generalised. Considering each two-band subspace separately, we have that all the patches with a half-integer Euler class must be connected (two-by-two) by a Dirac string, while the patches with an integer Euler class are not connected to any Dirac string.

As we will see, the advantage of introducing Dirac strings lies in the fact that it permits a direct visualisation of the global topological configuration of any multi-band nodal phase (within the plane that has $C_2T$ symmetry with $[C_2T]^2 = 1$). This will become clear below.

It is important to note that the precise trajectory of the Dirac string between two nodes of one gap can be freely modulated upon a local change of the gauge signs of the eigenvectors but this does not change the topological stability of the nodes, i.e. whether they can be annihilated (in which case we can assign them opposite frame charges) or not (in which case we can assign them the same frame charge).

There are other gauge freedoms when we consider the interaction of a Dirac string with the nodes of adjacent gaps. This is most efficiently summarised with a list of rules concerning the Dirac strings, which we now list as "Dirac strings rules"[11,15]:

i. All the linear nodes of a gap must connect two-by-two by a Dirac string, quadratic nodes (generically only existing at high-symmetry momenta) must be seen as the superposition of two linear nodes and are thus connected with themselves.

ii. The frame charge of a node (say $q_n$ for gap $n$) must flip sign ($q_n \rightarrow -q_n$) whenever it crosses a Dirac string of an adjacent gap [i.e. either gap $(n-1)$ or gap $(n+1)$]. Such a crossing can be the result of displacing the Dirac

string (as allowed by gauge freedom) while keeping the nodes fixed (which for a given phase are gauge invariant).

iii. Any couple of Dirac strings in the same gap can be recombined through the permutation of the end nodes. For example, if node 1 is connected with node 2 and node 3 is connected with node 4, we can also connect node 1 with node 3 and node 2 with node 4, or we can connect node 1 with node 4 and node 2 with node 3[15].

It should be noted that the Dirac string is a gauge object and hence cannot be physically observed. However, it is necessary for the definition of a global topological configuration of nodes, i.e. the consistent assignment of signed frame charges over the whole Brillouin zone. Such a picture has the merit of capturing the exhaustive topological structure of the phase, i.e. the (gauge invariant) relative stability of any pair of nodes is readily given by their assigned frame charges. Furthermore, any phase transition induced by the moving of the nodes over the Brillouin zone can be predicted from this global picture. The central mechanism of the phase transitions is the braiding of nodes, which we now introduce.

*Braiding processes.* One central mechanism of the topological phase transitions in $C_2T$ symmetric systems is the braiding of nodes[11,13,15]. We show it schematically in Fig. 6, which also illustrates the power of the pictorial approach enabled by the combination of the frame charges, the patch Euler class, and the Dirac strings.

We start in Fig. 6a with two pairs of nodes in adjacent gaps, each with opposite charges so that their patch Euler class (computed over the gray domain) is zero $\chi_n = \chi_{n+1} = 0$, indicating that both pairs can annihilate upon recombination along the Dirac strings. In Fig. 6b we choose different patches so that they contain the braiding trajectories of the nodes. In Fig. 6c we move the Dirac strings so that they both lie inside the patches. To achieve this, the full Dirac string (red) has to cross one open triangle, thereby flipping its frame charge. Similarly, the dotted Dirac string (blue) has to cross the open circle, also flipping its charge. In Fig. 6d we recombine the nodes within each gap together. The patch Euler classes are now nonzero, $\chi_n = \chi_{n+1} = 1$, due to the flip of the charges upon braiding, which reflects that the nodes are now stable, i.e. they cannot be annihilated within the patches.

Similarly, we can transfer a stable pair of nodes from one gap to an adjacent gap through the braiding of nodes, as shown in Fig. 6e–h.

*Determination of the topological configuration.* We can now determine the topological configuration of any multi-band nodal phase through the numerical calculation of patch Euler classes. We emphasise that the unsigned patch Euler class of a pair of nodes determines the relative stability of the nodes within the patch, which is a gauge invariant quantity.

First, we compute the absolute value of the patch Euler class of every single band crossings and of every pair of band crossings in the same gap, and repeat this for each gap. These constitute the gauge invariant quantities that constrain the assignment of signed frame charges and Dirac strings.

Once the patch Euler classes are determined, we build the global topological configuration of nodes like completing a puzzle. Choosing an initial pair of nodes, we assign a signed frame charge to each one under the constraint of the pair's patch Euler class, as well as the Dirac string connecting them. Then, we repeat the operation with every patch that overlaps with the previous one. Consequently, the assignments of the frame charges and the Dirac strings—still under the constraints of the computed patch Euler classes—are less arbitrary since the previous charges and Dirac strings have been fixed already. The complete topological configuration

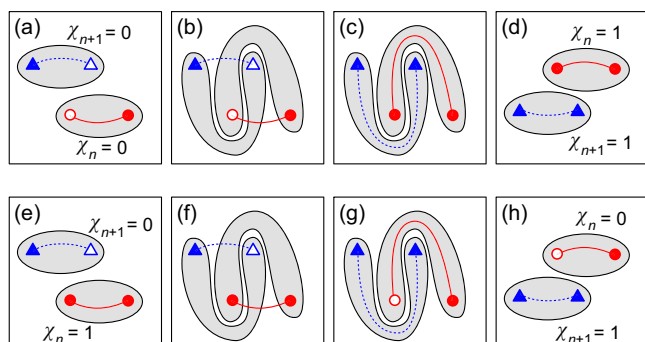

**Fig. 6 Schematic of the braiding processes. a–d** Braiding of adjacent nodes in gaps $\Delta_n$ and $\Delta_{n+1}$ leading to the obstruction of the annihilation of nodes. The outcome of the braiding is captured by the graphic representation of the non-Abelian frame charges $\{\pm q_n, \pm q_{n+1}\}$ together with their Dirac strings. The patch Euler classes capture the stability of the nodes upon recombination (braiding) within the chosen patches. **e–h** Transfer of a stable pair of nodes from one gap to an adjacent gap through the braiding of nodes.

then follows unambiguously as dictated by relative consistencies, i.e. consistency with the computed patch Euler classes and consistency with the previously fixed charges and Dirac strings.

We can now make use of all the conceptual tools exposed above (non-Abelian frame charges, patch Euler class, Dirac strings, and braiding processes) to obtain a pictorial representation of any nodal topological phase which is not only descriptive but also predictive. Indeed, from a known initial topological configuration we can predict the outcome of any phase transition happening through the displacement of the nodes induced by band inversions and braiding processes. We illustrate this in the main text with a detailed discussion of the band inversions in the phonon spectrum.

**Explicit braiding patterns via $C_6$ symmetry breaking.** One particularity of the braiding processes described in the main text is that they are collapsed to the high-symmetry points by the constrains of the hexagonal point symmetries of the crystal. This is the rationale for the formation of triply-degenerate nodes, i.e.

these are unavoidable whenever the nodes of one gap are pumped to an adjacent gap. Nevertheless, a more direct braiding pattern can be readily obtained by breaking the point symmetries responsible for the existence of the triply-degenerate nodes.

We illustrate this for the braiding process happening in group 1 of bands shown in Fig. 2. Thanks to the fact that the group 1 of bands is separated from the other bands, we can project three-band subspace on an effective three-band tight-binding model and reproduce the braiding process parametrised by a tuning variable $t \in [0, 1]$, with $t = 0$ and $t = 1$ corresponding to the electric field 0 eV/Å and 1.8 eV/Å, respectively. Then, by breaking the $C_6$ symmetry of the tight-binding model for $t \in [0 + \varepsilon, 1 - \varepsilon]$ ($\varepsilon \ll 1$), we obtain the braiding process in Fig. 7, where the red (blue) density plot corresponds to the nodes in gap 11 (gap 10). Figure 7a shows the process over the entire Brillouin zone, and Fig. 7b shows a zoom on the Γ point where the braiding between nodes of the adjacent gaps is clearly seen, leading to the transfer of the stable double nodes from gap 11 (red) to gap 10 (blue). We also show in Fig. 8 the detailed process of transfer of charge (frame charge −1, equivalently Euler class 1) in the vicinity of Γ mediated by the braiding. Finally, we

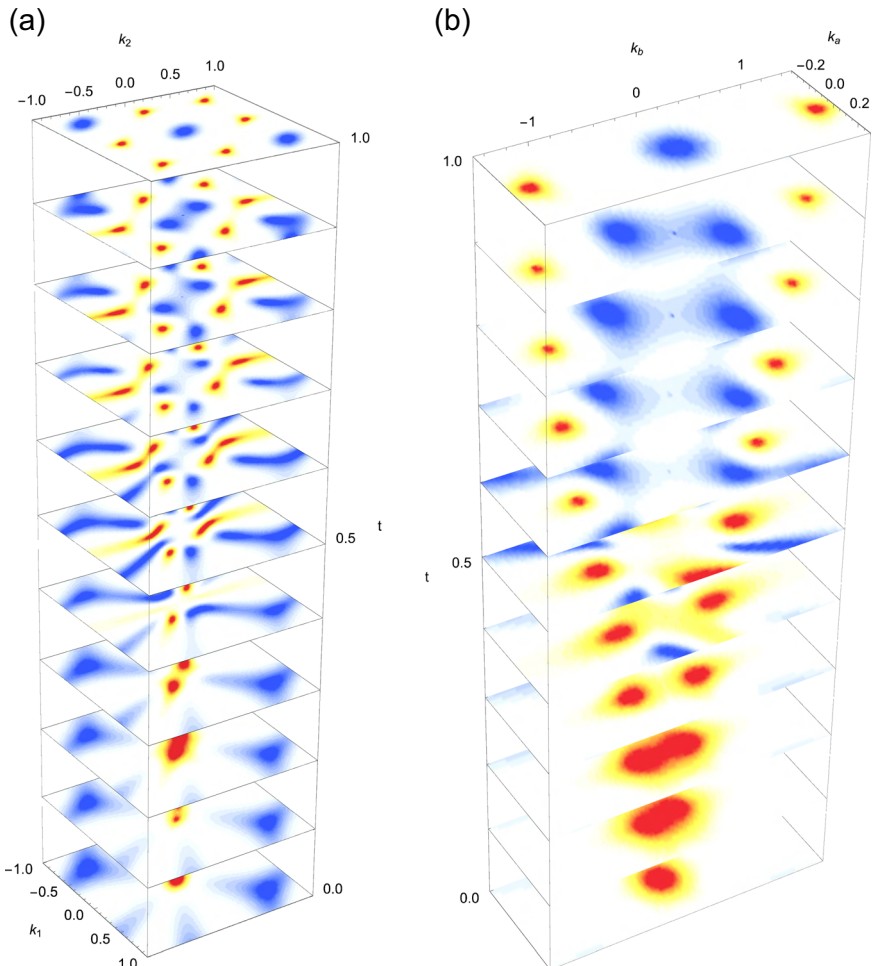

**Fig. 7 Braiding patterns via $C_6$ symmetry breaking.** Braiding within a three-band model of the group 1 of bands (see Fig. 2 in the main text) with broken $C_6$ symmetry **a** over the entire Brillouin zone $(k_1, k_2)/\pi \in [-1, 1]^2$, and **b** zooming around the Γ point. The red (blue) density plot corresponds to the nodes of gap 11 (gap 10), and the vertical axis $t \in [0, 1]$ corresponds to the electric field from 0 eV/Å to 1.8 eV/Å. See Fig. 8 showing the detailed transfer of charge due to the braiding in the vicinity of Γ.

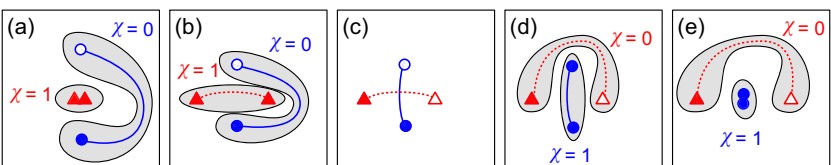

**Fig. 8 Schematic braiding process in the vicinity of Γ with broken $C_6$ symmetry.** The braiding processes lead to the transfer of charge (frame charge −1, equivalently Euler class 1) from gap $\Delta_{11}$ (red triangles) to gap $\Delta_{10}$ (blue circles). The successive panels **a**–**e** correspond to the braiding of Fig. 7b with $\varepsilon < t < 1 - \varepsilon$. The panels **b**–**d** have the same positions of nodes, while the Dirac strings are moved around and the charges are flipped accordingly.

note that the transfer of the simple nodes from $\Gamma$ to the K points converts them to stable double nodes, and this process is also visible in Fig. 7a.

**Algorithm for the non-Abelian charges of band-subspaces.** The computation of the non-Abelian SO($N$)-frame charges through the parallel transport of the frame and the lifting to the spin group Spin($N$) has been exposed in great details in ref. [10], see also ref. [112,114,115] on the computation of the generator of $\pi_1[\text{SO}(N)] = \mathbb{Z}_2$ as the geometric phase of the parallel transported orthogonal frame. While we mainly follow the derivation of ref. [10], we introduce here an algorithm that avoids the use of the Baker-Campbell-Hausdorff formula and generalises the computation of non-Abelian frame charges to partial frames of rank $M \le N$.

The algorithm we present here allows the computation of non-Abelian charges from *partial frames*, i.e. from a subset of the Bloch eigenvectors of the band-subspace. More precisely, given a band-subspace that is separated from the other bands by an frequency gap above and below over a patch $\mathcal{D}$ bounded by the base loop $l = \partial\mathcal{D}$, we obtain the non-Abelian frame charge over $l$ independently of the nodes located outside the selected band-subspace. Since there is no algorithm for the wannierisation of phonon bands onto effective few-band tight-binding models, and given that the definition of frame charges from the total frame becomes quickly cumbersome with an increasing total number of bands $N$[10] (e.g. $N = 21$ for silicate leading to 210 Dirac matrices needed to form a basis of Spin(21)), our algorithm brings a great simplification of the computation of the non-Abelian frame charges in real materials.

Importantly, we have used these results to corroborate the topological configurations derived from the patch Euler classes. When there are unavoidable $M(>2)$-fold band crossings (e.g. this happens in crystalline systems with a cubic point group), the Euler class cannot be used, and the direct computation of partial-frame charges become necessary.

We note finally that the algorithm remains valid in any other type of band structures, e.g. in electronic band structures.

*The algorithm.* In a system with a total number of bands $N$, we consider a group of $M(\le N)$ bands which are isolated from all other bands by an frequency gap above and below over a patch bounded by the loop $l$. We define the partial frame of rank $M$ for this group of bands as a function of a point of the base loop $l$,

$$R_{n+1,n+M}(\mathbf{k})|_{\mathbf{k}\in l} = (u_{n+1}(\mathbf{k})u_{n+2}(\mathbf{k})\cdots u_{n+M}(\mathbf{k}))_{\mathbf{k}\in l} \in \mathbb{R}^N \times \mathbb{R}^M, \quad (2)$$

where $\{u_i\}_{i=n+1,\dots,n+M}$ are the Bloch eigenvectors corresponding to the eigen-frequency $\{E_i\}_{i=n+1,\dots,n+M}$ that are ordered as $E_n(\mathbf{k}) < E_{n+1}(\mathbf{k}) \le \cdots \le E_{n+M}(\mathbf{k}) < E_{n+M+1}(\mathbf{k})$ for all $\mathbf{k} \in l$. We assume that $l$ is a contractible base loop in the Brillouin zone (i.e. it is not crossing the periodic Brillouin zone).

Discretising the base loop into $N_l$ points, i.e. $l \cong [\mathbf{k}_0, \mathbf{k}_1, \cdots, \mathbf{k}_{N_l} \equiv \mathbf{k}_0]$, we form the list of parallel transported partial-frames. Starting with the partial-frame at the initial base point, $R_{\mathbf{k}_0}$, which we will keep fixed, we first define the projection with the partial-frame at the next point of the base loop, i.e. $F_{\mathbf{k}_1} = R_{\mathbf{k}_1}^T R_{\mathbf{k}_0} \in \text{O}(M)$. We then redefine the partial-frame at $\mathbf{k}_1$ by smoothing its gauge phase through the substitution $R_{\mathbf{k}_1} \to \widetilde{R}_{\mathbf{k}_1} = R_{\mathbf{k}_1}\text{diag}[\text{sign}(F_{\mathbf{k}_1,11}), \dots, \text{sign}(F_{\mathbf{k}_1,MM})]$. This gives us the oriented projection $\widetilde{F}_{\mathbf{k}_1} = \widetilde{R}_{\mathbf{k}_1}^T R_{\mathbf{k}_0} \in \text{SO}(M)$. Proceeding iteratively for all the points of the discretised loop, we obtain $N_l$ oriented projection matrices of the parallel-transported partial-frames, i.e. $\{\widetilde{F}_{\mathbf{k}_i} = \widetilde{R}_{\mathbf{k}_i}^T \widetilde{R}_{\mathbf{k}_{i-1}}\}_{i=1,\dots,N_l}$.

When the mesh of the discretisation is fine enough, each parallel-transported projection matrix $\widetilde{F}_{\mathbf{k}_i}$ is close to the identity matrix in SO($M$), which implies that the matrix log function, $\log : \text{SO}(M) \to \mathfrak{so}(M)$, is one-to-one[116]. We thus get the decomposition of the projection matrices into their $\mathfrak{so}(M)$ Lie-algebra components, i.e. $A_{\mathbf{k}_i} = \log\widetilde{F}_{\mathbf{k}_i} = \sum_{ab\in I}\theta_{ab}(\mathbf{k}_i)L_{ab}$ with the $M(M-1)/2$ skew-symmetric $(M \times M)$ matrices[10] $[L_{ab}]_{ij} = -\delta_{ai}\delta_{bj} + \delta_{aj}\delta_{bi}$ that form a basis of $\mathfrak{so}(M)$ and which we label by $ab \in I = \{(a,b)|1 \le a < b \le M\}$, where $|I| = M(M-1)/2$.

Once the Lie-algebra decomposition of the projection matrices is known, we lift them into the (universal) double covering group of SO($M$), that is the spin group Spin($M$)[10]. The lift is most easily done at the level of the Lie algebras, i.e. $\mathfrak{so}(M) \to \mathfrak{spin}(M)$. Indeed, the two Lie algebras are isomorphic[116] such that there is a one-to-one correspondence between their matrix representations, i.e. $L_{ab} \leftrightarrow t_{ab}$, where the matrices $\{t_{ab}\}_{ab\in I}$ form a basis of $\mathfrak{spin}(M)$. The $t_{ab}$ matrices are most conveniently defined through $t_{ab} = -\left\{\begin{matrix}1\\i\end{matrix}\right\}\frac{1}{4}[\Gamma_a, \Gamma_b]$, where $\{\Gamma_a\}_{a=1,\dots,M}$ are the $2^{\lfloor M/2 \rfloor} \times 2^{\lfloor M/2 \rfloor}$ gamma matrices which anti-commute one with another[10]. The lifting of projection matrices to Spin($M$) is readily given by $\overline{F}_{\mathbf{k}_i} = \exp[\sum_{ab\in I}\theta_{ab}(\mathbf{k}_i)t_{ab}]$[10].

We then obtain the accumulated projection matrix through the path-ordered product

$$\Delta\overline{F}_{\mathbf{k}_j} = \overline{F}_{\mathbf{k}_j} \cdot \overline{F}_{\mathbf{k}_{j-1}}\cdots\overline{F}_{\mathbf{k}_1} \cdot \overline{F}_{\mathbf{k}_0}, \text{ with } \overline{F}_{\mathbf{k}_0} = \mathbb{1}, \text{ and for } j = 0,\dots,N_l, \quad (3)$$

and the matrix log gives its $\mathfrak{spin}(M)$-decomposition

$$\Delta\overline{A}_{\mathbf{k}_i} = \log\Delta\overline{F}_{\mathbf{k}_i} = \sum_{ab\in I}\gamma_{ab}(\mathbf{k}_i)t_{ab}. \quad (4)$$

From the above expression, we define the accumulated geometric phases per spin-component as $\{\gamma_{ab}(\mathbf{k}_i)\}_{ab\in I}$. Forgetting the discretisation, we write the geometric phases acquired by rotating partial-frame over a base loop as

$$\gamma(\mathbf{k})\big|_{\mathbf{k}\in l} = \left(\gamma_{12}(\mathbf{k}), \cdots, \gamma_{(M-1)M}(\mathbf{k})\right)\big|_{\mathbf{k}\in l}. \quad (5)$$

Fixing a reference point $\mathbf{k}_{\text{ref}}$ inside the open disc $D$ that is bounded by the base loop $l$, i.e. $\partial\overline{D} = l$, we parametrise a point of the base loop, $\mathbf{k} \in l$, by a polar angle $\theta_{\text{ref}}$ through $(\cos\theta_{\text{ref}}, \sin\theta_{\text{ref}}) = (\mathbf{k} - \mathbf{k}_{\text{ref}})/|\mathbf{k} - \mathbf{k}_{\text{ref}}|$. The geometric phases over the complete disc are then defined as $\gamma[l] = \gamma(\theta_{\text{ref}} = 2\pi)$. Since the base loop is closed (i.e. it is contractible in the Brillouin zone) we have the following boundary condition for the parallel-transported partial-frame

$$\widetilde{R}(\theta_{\text{ref}} = 2\pi) = \widetilde{R}(\theta_{\text{ref}} = 0) \cdot \text{diag}(g_{n+1}, \cdots, g_{n+M}), \quad (6)$$

where $\{g_{n+i}\}_{i=1,\dots,M}$ are gauge signs $\pm 1$. It can be shown (following an argument along the line of ref. [114] after a lifting to the spin group and spin algebra, this will be shown elsewhere), that the above boundary condition leads to the quantisation of the geometric phases after a complete cycle on the base loop. I.e. writing $\gamma_{ab}(\mathbf{k} \in l) = \gamma_{ab}(\theta_{\text{ref}})$, the geometric phases are quantised by

$$\gamma_{ab}(2\pi) \in \{0, \pm\pi, 2\pi\} \text{ for all } ab \in I. \quad (7)$$

Defining the $M(M-1)/2$ non-Abelian charges

$$\{q_{ab} = e^{\gamma_{ab}(2\pi)t_{ab}}\}_{ab\in I}, \quad (8)$$

the matrix representation of the double group $\overline{\text{P}}_M$ is given by[10]

$$\overline{\text{P}}_M = \bigcup_{n_{ab}\in\{0,1\}}\left\{\pm\prod_{ab\in I}q_{ab}^{n_{ab}}\right\}. \quad (9)$$

Any pair of non-Abelian charges with a single common index anti-commutes, i.e. $\{q_{ab}, q_{bc}\} = 0$ with $c \neq a$, while any pair with no common index commutes, i.e. $[q_{ab}, q_{cd}] = 0$ for $a \neq b \neq c \neq d$. Furthermore, any pair with a single common index satisfies the contraction $q_{ab}q_{bc} = q_{ac}$ (this condition can actually be used to set the definition of the $\{t_{ab}\}_{ab\in I}$ matrices). Taking into account these product rules, the group $\overline{\text{P}}_M$ can be decomposed into $2^{M-1} + 1$ distinct conjugacy classes for $M$ odd and $2^{M-1} + 2$ for $M$ even[10]. We conclude by noting that, while the sign of each non-Abelian charge $q_{ab}(\neq \mathbb{1}_{2^{\lfloor M/2 \rfloor}}, -\mathbb{1}_{2^{\lfloor M/2 \rfloor}})$ depends on the choice of the gauge signs at the base point of the loop (i.e. in the definition of $R_0$), the relative stability of any pair of nodes within the same gap is gauge invariant[10,13] and is indicated by the charges $\mathbb{1}_{2^{\lfloor M/2 \rfloor}}$ (when the pair can annihilate) and $-\mathbb{1}_{2^{\lfloor M/2 \rfloor}}$ (when the pair cannot annihilate). Alternatively, this relative stability can be characterised by the patch Euler class.

## Data availability

The data that support the findings of this study are available from the corresponding author upon reasonable request.

## Code availability

VASP for DFT calculations is available at https://www.vasp.at. PHONOPY for phonon calculations is available at https://phonopy.github.io/phonopy. WANNIERTOOLS for computing the band crossings and edge states is available at https://github.com/quanshengwu/wannier_tools, with the phonon tight-binding Hamiltonian generated by a python tool developed by Dr. Changming Yue. The MATHEMATICA NOTEBOOK for the Euler class calculations is available at ref. [86], as described in ref. [13]. The calculated phonon eigenvectors are rotated to a real basis as input, as described in ref. [117].

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

## Acknowledgements

B.P., B.M., and R.-J.S. acknowledge funding from the Winton Programme for the Physics of Sustainability. B.M. also acknowledges support from the Gianna Angelopoulos Programme for Science, Technology, and Innovation. R.-J.S. also acknowledges funding from the Marie Skłodowska-Curie programme under EC Grant No. 842901 and from Trinity College at the University of Cambridge. The calculations were performed using resources provided by the Cambridge Tier-2 system, operated by the University of Cambridge Research Computing Service (www.hpc.cam.ac.uk) and funded by EPSRC Tier-2 capital grant EP/P020259/1, as well as with computational support from the U.K. Materials and Molecular Modelling Hub, which is partially funded by EPSRC (EP/P020194), for which access is obtained via the UKCP consortium and funded by EPSRC grant ref. EP/P022561/1.

## Author contributions

All authors contributed extensively to all aspects of the work presented in this paper. All authors designed the project. B.P. and B.M. performed the ab-initio calculations and A.B. and R.J.S. provided the theoretical description. All authors contributed in the writing of the manuscript.

## Competing interests

The authors declare no competing interests.
