## [Peer Review File · Nature Communications]

REVIEWER COMMENTS

Reviewer #1 (Remarks to the Author):

The authors study the non-Abelian topological physics of phonon in silicate. The non-Abelian topological charge is an emerging new field in physics. In this context, this paper has a potentiality to be published in Nature Communications. However, the current manuscript is too concise to present such a new physics. The authors should substantially expand the Supplementary Information. The supplementary information should be newly written for this manuscript. Indeed, there are many overlaps in the present supplementary information copied from other papers of their own.

Here, I list some questions on the results of the main text, to which the author should carefully address.

1)

I do not agree that the non-Abelian nature is observed by Raman spectra. Although I agree that the Raman spectra can present each snap shot phase of phonon spectra, they do not reflect the non-Abelian topology. For example, how can we observe a quaternion algebra by the Raman spectra? The quaternion algebra indicates that the order of the braiding is critical. Is it possible to observe such ordering of the braiding by the Raman spectra?

2)

The derivation of non-Abelian topological charges is too weak. The readers cannot follow the derivation as it is. Because this is an essential part of their work, it should be detailed in Supplementary Information.

3)

Is it possible to construct a tight-binding model for the system? Only the first-principle results are not easy to understand. It is highly desirable to construct a tight-binding model, which will be done by using the maximally localized Wannier function method.

4)

What is the characteristic feature of the edge states in the non-Abelian topological phase? Fig.2 seems to be a usual edge states for an Abelian topological phase. What is the multi-band nature of these edge states?

5)

The terminology "braid" has a strict sense implying the braid algebra. On the other hand, it is trivial that quaternion algebra does not satisfy the braid relation. The authors should clarify the difference between the braid algebra and the quaternion algebra.

For example, the authors should show the difference between $ij=k$ and $ji=-k$. Without it, one cannot determine whether the system is truly non-Abelian or not. How can we differentiate them experimentally?

6)

The authors should show 3D pictures on the "braiding" process. Only illustrations in 2D pictures are misleading enough since we can not see how the braiding occurs.

7)

Some of the authors have written a related paper Ref.15 and uploaded it in arXiv, which also deals with non-Abelian topological physics of phonon. The authors should clarify the novelty and importance of the present work by comparing with Ref.15.

8)

The authors should clarify the relation between the quaternion charge and the Euler number. Some papers discuss that the Euler number is not sufficient to characterize the non-Abelian topology. However, this paper gives an impression that the Euler number is enough.

9)

The patch Euler number is not clearly explained. For example, the authors should explicitly show the region of the patch. The authors should show the integral and the integrand of the Euler number.

10)

Is it possible to observe the Dirac string experimentally in the present system?

11)

PT or C2T symmetry is necessary for the quaternion charge. However, it is not clear which symmetry exists in the present system. The authors should clarify why such a symmetry exists in silicate.

12)

Is the system robust as long as PT or C2T symmetry is preserved?

The authors should discuss the effect of impurities and randomness in order to clarify the topological protection. Without it, one cannot determine whether the system is truly topologically protected or accidentally protected.

13)

In other papers on non-Abelian topological charges, they use the spin Berry-Wilczek-Zee phase instead of the patch Euler number. The authors should discuss the relevance between them.

Reviewer #2 (Remarks to the Author):

In the manuscript, the authors propose that multi-gap topologies and related node conversion processes can be observed in the phonon spectra of monolayer silicates by applying an electric field and epitaxial strain. The authors made significant effort to describe the complicated node conversion processes in terms of creation and annihilation of topological nodes carrying nonabelian frame charges, and provided an elegant theory for the evolution of phonon band structure under strain and electric field. It is also proposed that a part of the node conversion processes can be monitored by observing the Raman spectra.

I think that the theoretical observation of the node conversion associated with the non-abelian frame charge and multi-gap band topology in phonon spectra is interesting, and the theoretical analysis is rigorous and reliable. I also think that the manuscript appeared in a timely manner as there is a growing interest in the non-abelian topological charges of nodes in recent studies.

However, I also have several concerns about the novelty of this paper, especially because (i) feasibility of the proposed node conversion in real material is not clear, (ii) there is no thorough analysis of the edge states, (iii) the manuscript is still quite technical and theoretical description is complicated.

Below I give several questions and comments, which should be properly addressed by the authors, before any decision can be made about the manuscript.

(1) In the section about phonon band structure, the low energy phonon bands, from band 1 to 9, are not considered. What happens in these phonon bands when electric field and strain are applied?

(2) The stability of the lattice structure and the symmetry of monolayer silicate under strain and electric field should be more carefully analyzed at the level of first-principles calculations, including the low energy phonon bands. Does the material maintain the same structure and symmetry under 7 %

strain and 2.0 eV/Å electric field, or even under higher strain and electric field? What supports the stability? What is the stability limit?

(3) Is it possible to apply 7 % strain and 2.0 eV/Å electric field simultaneously in experiments? The experimentally accessible region of strain and electric field should be clearly mentioned.

(4) In the section about "Braiding in group 1", it is stated that "Applying an electric field increases the frequency of the non-degenerate band B₁₀ at Gamma while reducing the frequency of the degenerate bands B_{11,12}". What is the physical reason for the change of the energy ordering under electric field? Is it possible to explain it physically by considering the deformation pattern of the relevant phonon modes?

(5) Although the authors claimed that braiding of nodes can be controlled, what is actually happening in the system is complicated conversion processes of many nodes simultaneously. In the usual description of braiding in real space, one can imagine two point-like objects whose positions can be interchanged. Can it be possible to perform such a braiding of individual node in the proposed phonon system, instead of the simultaneous conversion of many nodes? If such a process is impossible, I am wondering how useful the momentum space braiding is, compared to the relevant real space counterpart.

(6) To describe the linear and quadratic nodes, the authors used the convention that linear nodes are represented by small symbols and quadratic nodes by large symbols. But it seems to be better to use another convention for clarity.

(7) In the section about edge states, the authors mentioned that "we take a spectator view by directly visualizing the edge states but without addressing the fundamental mechanisms behind them". But the proper Zak phase analysis for the proposed phonon band structure should be performed. What is the origin of the edge arc connecting two adjacent nodes at the neighboring Gamma points on the (100) edge? Proper explanation for the other edge spectra also should be provided.

(8) The Raman spectrum can provide only a part of the information about the node conversion. This point should be clearly mentioned in the manuscript.

(9) The manuscript is still quite technical and complicated, which disturb the readability of the manuscript.

Reviewer #3 (Remarks to the Author):

In this work Peng et al study the non-Abelian topological charges in a layered silicates system, via straining and/or an applied DC bias. They theoretically calculate the band structure of layered silicates, which consists of three groups of bands. Focusing on the three-band in the lowest group, the authors show that straining can reduce the band energy difference close to the gamma point. By further applying a voltage, the band crossing can be induced, leading to creation of additional band nodes between the three bands. Varying the strain and/or voltage drives the motion of the band nodes in the Brillouin zone. The induced braiding behaviour between nodes of different bands are studied in theory. The author further present the simulation results of the Raman spectra at different applied strains and voltages, and claim that the braiding behaviour can be directly observed in experiment via the evolution of Raman spectra.

The work proposes a realistic system for observing the interesting braiding band structure for phonons. The physics is interesting but the presentation of the paper is far from being accessible to readers. Before I can make a recommendation, the authors need to address the following points.

1. How realistic is the proposed strain level, e.g 5% and 7%?
2. The presentation is very unclear, and this makes it very difficult to follow the paper. For example, Fig. 2b was not explained in detail. How is the Dirac string configuration obtained? Are the configurations of the Dirac string plotted in Fig. 2b unique?
3. The Raman signal only traces the shift of the bands at Gamma point, while the key features of

braiding in Fig. 2-4 are manifested in the motion of the degeneracy nodes away from the Gamma point. So the claim "the braiding processes described above can be directly observed experimentally by following the evolution of the Raman spectrum of the material" is quite overstated.

Reviewer #1 (Remarks to the Author):

The authors study the non-Abelian topological physics of phonon in silicate. The non-Abelian topological charge is an emerging new field in physics. In this context, this paper has a potentiality to be published in Nature Communications. However, the current manuscript is too concise to present such a new physics. The authors should substantially expand the Supplementary Information. The supplementary information should be newly written for this manuscript. Indeed, there are many overlaps in the present supplementary information copied from other papers of their own.

Here, I list some questions on the results of the main text, to which the authors should carefully address.

Answer: We would like to thank the reviewers for their work in reviewing our manuscript and for their detailed comments.

Regarding the comment that “there are many overlaps ... copied from other papers of their own”, we have re-written the Method section and also added our new results on partial frame charges.

1) I do not agree that the non-Abelian nature is observed by Raman spectra. Although I agree that the Raman spectra can present each snap shot phase of phonon spectra, they do not reflect the non-Abelian topology. For example, how can we observe a quaternion algebra by the Raman spectra? The quaternion algebra indicates that the order of the braiding is critical. Is it possible to observe such ordering of the braiding by the Raman spectra?

Answer: We thank the reviewer for pointing this out, and we agree with the reviewer that, taken out of context, the Raman spectrum as such only provides information on band inversion at Γ , and does not in itself reflect quantities like the quaternion algebra. The reason we think the Raman spectrum results are important is that they precisely allow us to *track* the band inversion processes which, because of the non-Abelian nature of the system, indicate that underlying braiding processes and inter-gap transfer of nodes are taking place. The band inversion itself thus carries the key information of the number of stable nodes pumped into the different gaps. Providing the direct signature of the band inversion in our context of phonon band structures is therefore crucial. We do hope that the reviewer shares our opinion that the evolution of the Raman spectrum is a new and interesting avenue to explore, especially in regard to phonon band topology, and that by keeping track of the band inversions we do obtain information of the braiding process, hence showing the potential of these observations.

More generally, we note that our results are in line with the wider topology research field, in which experimental observations cannot measure the topological invariant directly. For example, electronic topological semimetals or insulators are typically probed with ARPES, revealing relevant band crossings, but again, these by themselves are not sufficient to capture the fundamental nature of band topology. Instead, what is done in all these cases (and what we are also doing in our work) is to interpret experimental observations in the language of topology by supporting experiments with the microscopic theory. Considering the particular case of the multi-gap non-Abelian topology, we would like to point out that even when the Bloch eigenstates are measured, theoretical insights are necessary to interpret any non-Abelian charge. (Indeed, an initial gauge choice, a parallel transport, and the lifting of the $so(N)$ -connection into the spin algebra are all needed for the unambiguous determination of the non-Abelian charges in 1D.) What our results add is that, once we understand the topological features microscopically, we show that we can use the evolution of the Raman spectrum to track the band inversion, hence thereby effectively tracking the associated braiding and node-transfer processes.

We have clarified this as follows,

(Abstract)

Finally, we propose that *the band inversion processes at the Γ point* can be tracked by following the evolution of the Raman spectrum, providing a clear signature for the experimental verification of *the band inversion accompanied by the braiding process*.

...

(Introduction)

We further show that the evolution of Raman peaks can be used to track the band inversions and *the accompanying braiding process*, providing a clear experimental signature to identify multi-gap topology in real materials.

...

(Experimental signature: Raman spectra)

We finally propose that the *band inversion* processes described above can be directly observed experimentally by following the evolution of the Raman spectrum of the material.

...

These calculations show that Raman spectroscopy can be a promising tool for characterizing *the band inversion processes and the accompanying non-Abelian braiding of phonons* in 2D materials such as monolayer silicate.

2) The derivation of non-Abelian topological charges is too weak. The readers cannot follow the derivation as it is. Because this is an essential part of their work, it should be detailed in Supplementary Information.

Answer: We thank the reviewer for pointing this out. We have taken the opportunity of this revised version to include a new algorithm, presented in detail in Methods, that allows the computation of the non-Abelian frame charges within any *partial frame*. So far, the literature has only considered the computation of frame charges where *all the eigenstates* of the Bloch Hamiltonian must be taken into account. This represents an important limitation in the context of band structures in real materials where a large number of bands can be connected together. (While this limitation in particular can be avoided in meta-materials by design, with a band-subspace of interest that is well separated in energy from the other modes, this is not the case with the phonon band structure of solid state materials which are “designed” by nature.) For instance, a node with a frame charge of -1 appearing in a given band-subspace may be trivialized by a similar node appearing in the same band-subspace above or below in energy. In this new version we have corroborated the non-Abelian charges implied by the patch Euler class through a direct computation of the partial-frame charges (see also our reply to questions 8) and 13). This also demonstrates the advantage of the patch Euler class, which, using only two bands at a time by definition and by taking a value in \mathbb{Z} (while the non-Abelian charges can only distinguish the parity of the number of identical frame charges), gives the finest topological resolution of the band structure. Then, our algorithm for the partial-frame charges can be used whenever an $N > 2$ -fold degenerated band crossing appears (since with $N > 2$, the patch Euler class cannot be used). On a more technical note, we also point out that our algorithm avoids the use of the Baker-Campbell-Hausdrof formula (as was originally done in Ref.[10]) which becomes cumbersome for accurate results when the Lie group element is far from the identity, as such a situation precisely appears for the -1 (parital) frame charge.

3) Is it possible to construct a tight-binding model for the system? Only the first-principle results are not easy to understand. It is highly desirable to construct a tight-binding model, which will be done by using the maximally localized Wannier function method.

Answer: One advantage of our phonon calculations is that we can obtain the force constants directly from first principles, which are the second derivatives of the total energy with respect to two atomic displacements. Therefore, the force constants can be viewed as a numerically exact tight-binding Hamiltonian that includes all the hopping terms within the supercell. Crucially, thanks to our new algorithm for non-Abelian charges based on partial frames, whenever there is an M -band subspace containing the nodes of interest, we now can directly compute the effective $O(M)/O(1)^M$ -non-Abelian charges. There is therefore no need anymore to derive effective M -band tight binding models. Furthermore, in practise the use of the numerical Bloch eigenvectors obtained for the phonon spectrum from first-principle results is as straightforward as it is with a tight-binding model. Therefore, also in terms of complexity, there is no need to rely on an effective tight-binding model that would only be an approximation of the data we are using.

4) What is the characteristic feature of the edge states in the non-Abelian topological phase? Fig. 2 seems to be a usual edge states for an Abelian topological phase. What is the multi-band nature of these edge states?

Answer: We thank the reviewer for pointing this out. We have performed Zak phase calculations and found that for the three-band braiding the Zak phase for each gap can describe the edge states of the three bands effectively whereas for the four-band braiding the Zak phase fails to provide a consistent picture. We have clarified these points, as follows,

Edge states

We also study the evolution of topological edge states with varying electric field by calculating the surface local density of states (LDOS) from the imaginary part of the surface Green's function.⁷¹ Figure 1(c) shows the edge states of 7% strained silicate. On the (100) edge (i.e. the zigzag direction for the Si atoms), there is an edge arc connecting two adjacent nodes at the neighbouring Γ points. Upon increasing electric field, new band crossing points appear around Γ . At 1.8 eV/Å, there are two clear projections of the new Dirac cones on the (100) edge around 17.36 THz. We calculate the Zak phase γ (i.e. the Berry phase along a non-contractible path of the Brillouin zone which is here quantised to $\{0, \pi\}$ by the C_2T symmetry) in gap Δ_{10} and Δ_{11} at 0.0 and 1.8 eV/Å. As shown in Fig. 1(c), a Zak phase of zero corresponds to the emergence of the edge states, while $\gamma = \pi$ leads to vanishing edge states. This indicates that the band Wannier states (the Wannier states are the Fourier transform of the Bloch eigenstates) have their center (the “band centers”) shifted from the center of the atomic Wannier states (the “atomic centers”), leading to a charge anomaly and, following, the localisation of an edge state. Furthermore, the fact that the edge states appear when the Zak phase is zero simply means that the band centers occupy the center of the unit cell, while the atomic centers lie on the boundary of the unit cell.^{15,72} Indeed, the \mathbb{Z}_2 quantised Berry phase is a good quantum number in 1D, which indicates the band center (i.e. its Wyckoff position).⁷³

We can actually predict the Zak phase for the given edge termination from the the bulk topology. The Zak phase at a fixed gap is given (modulo 2π) by the parity of the number of Dirac strings that are crossed by the straight path of integration that is perpendicular to the edge axis, i.e. for the (100) edge these are the paths $l_{k_{\parallel}} \in \{k_{\perp}(b_1 - b_2) + k_{\parallel}(b_1/2 + b_2/2) | k_{\perp} \in [0, 1]\}$ at a fixed $k_{\parallel} \in [0, 1]$, with k_{\parallel} the coordinate of the horizontal axis in Fig. 1(c) and Fig. 2(c).¹⁵

Figure 2(c) shows the topological edge states of 5% strained silicate on the (100) edge. Besides the edge arc that connects two adjacent nodes at the neighbouring Γ points around 26.45 THz, there is also a new edge arc connecting two adjacent nodes around 25.90 THz located at neighbouring K points with 2D irrep K_3 . Under an electric field of 0.9 eV/Å, extra edge arcs emerge near $\bar{\Gamma}$ at 25.97 THz. The arc connecting K_3 moves downwards upon increasing electric field and is robust. For the four-band braiding

Figure 1. The corresponding phonon edge states on the (100) edge are shown in (c), with the $\{0, \pi\}$ -quantised Zak phases $\{\gamma_n\}_{n=10,11}$ for gaps $\{\Delta_n\}_{n=10,11}$ indicated by the arrows.

processes, the Zak phase argument fails, as the Zak phases γ_{14-17} in Fig. 2(c) do not show a consistent behaviour. We stress that, apart from effective Zak phase diagnoses,^{15,74} the full multi-gap bulk-boundary correspondence remains an open question. Hence, we take here a “spectator” view by directly visualising the edge states but without addressing the fundamental mechanisms behind them.

Figure 2. The corresponding phonon edge states on the (100) edge are shown in (c), with the $\{0, \pi\}$ -quantised Zak phases $\{\gamma_n\}_{n=14,\dots,17}$ for the gaps $\{\Delta_n\}_{n=14,\dots,17}$ indicated by the arrows.

5) The terminology “braid” has a strict sense implying the braid algebra. On the other hand, it is trivial that quaternion algebra does not satisfy the braid relation. The authors should clarify the difference between the braid algebra and the quaternion algebra. For example, the authors should show the difference between $ij = k$ and $ji = -k$. Without it, one cannot determine whether the system is truly non-Abelian or not. How can we differentiate them experimentally?

Answer: We respectfully disagree here with the referee. The notion of the “braiding” of nodal points (lines) as being associated with the conversion of their non-Abelian frame charges has been introduced very precisely in Ref. 1 and 2. Namely, by fixing a base point and an oriented base loops over which we compute the frame charges (this is a necessary assumption by the very definition of the first homotopy group), whenever a nodal point with a frame charge q_n has its charge converted to $-q_n$, it is braided around a nodal point of an adjacent gap, see Fig.1 in Ref. 2, see also Fig.S9 and Eq.S43 of Ref. 1 where the correspondence between the quaternion algebra and the braid trajectory of the node is shown explicitly.

From a more mathematical viewpoint, the quaternion group is isomorphic to the pure braid group of two strands on the projective plane, i.e. $\mathbb{Q}_8 \cong P_2(\mathbb{R}P^2)$.

Addressing now the question of the experimental manifestation of the charge conversion through braiding, we explain in Methods that the relative stability of a pair of nodes is directly indicated by the frame charge ± 1 over the loop that encircles both nodes. The stability of two nodes, i.e. the obstruction to their annihilation, is a relative notion and, contrary to the absolute charge of a single node, is gauge invariant. This $q \in \mathbb{Z}_2$ notion of relative stability is then generalized to a $2\chi \in \mathbb{N}$ classification by the patch Euler class, i.e. while the frame charge can only capture the parity of the number of stable pairs of nodes, the Euler class can count the stable accumulation of an arbitrary (odd or even) number of nodes (for an odd number of stable nodes, the Euler class χ takes a half-integer value, while it is even for an even number of nodes). As explained in Methods, the relative stability of nodes can only be changed through a braiding process, during which stable nodes can be transferred (pumped) from one gap to the adjacent gaps. We explain that this is necessarily happening during certain band inversions. We thus can say that our work is primarily concerned with the physical manifestations of the braiding of nodes and the conversion of their non-Abelian charges, leading to changes in their relative stability (see also our reply to the next question). We furthermore show that the merging of nodes at high-symmetry points affect the dispersion which is an indication of the conversion of their patch Euler class and frame charge.

In the new version, we have clarified these points through the whole manuscript and this is also reflected in the methods section that has been substantially extended. Furthermore, there is now a new section in the main text on

The Euler class-dispersion relation

We make the general observation that for any single band crossing at a momentum k_0 , the patch Euler class, $\chi[k_0]$, determines the lower bound of the degree of the dispersion of the Bloch eigenvalues, $\lambda(k)$, at the band crossing: defining the order of the leading term in a Taylor expansion of the Bloch eigenvalues at k_0 as α , we get $2\chi[k_0] \leq \alpha$.¹⁵ For instance, a single node with Euler class 0.5 must exhibit a dispersion at least linear ($2\chi = 1 \leq \alpha$). We call this the Euler class-dispersion relation. In that regard it is crucial to remember that the eigenvalues of the dynamical matrix are frequencies squared ($\lambda = \omega^2$) while the phonon band structures are plotted as a function of frequency ($\omega = \sqrt{\lambda}$). Defining by β the order of the leading term of a Taylor expansion of the phonon frequencies at a band crossing, the Euler class-dispersion relation thus gives $\chi[k_0] \leq \alpha/2 = \beta$. Very interestingly, except for the lowest phonon bands at Γ ,²⁶ the dispersion of all the other band crossings always exhibits an order strictly higher than their Euler class lower bound, i.e. we find $2\chi[k_0] \leq \beta$. This implies that the hopping terms of the wannierised dynamical matrix are dominated by long-range hopping processes, contrary to electronic systems where the short-range hopping processes are strongly dominant due to screening.

6) The authors should show 3D pictures on the “braiding” process. Only illustrations in 2D pictures are misleading enough since we can not see how the braiding occurs.

Answer: We thank the reviewer for pointing this out. We have added a new section in methods which addresses this point explicitly with the title *Explicit braiding patterns via C_6 symmetry breaking*. There, we use an effective three-band tight-binding model that corresponds to the group 1 of bands and we reveal the 3D braid trajectory that must take place during the band inversion at the Γ point, as shown in Fig. 3. Figure 3(a) shows the 3D braiding over the whole Brillouin zone, and Fig. 3(b) makes a zoom on the Γ point. We obtain the braid trajectory by breaking the C_6 symmetry while we preserve the C_2T symmetry. As we explain in the new version, in the silicate system the relative braid trajectories are collapsed on the high-symmetry points as a consequence of the crystalline symmetry constraints.

Figure 3. Braiding within a three-band model of the group 1 of bands (see Fig. 2 in the main text) with broken C_6 symmetry (a) over the entire Brillouin zone $(k_1, k_2)/\pi \in [-1, 1]^2$, and (b) zooming around the Γ point. The red (blue) density plot corresponds to the nodes of gap 11 (gap 10), and the vertical axis $t \in [0, 1]$ corresponds to the electric field from 0 eV/\AA to 1.8 eV/\AA .

In particular, in Fig. 3 the vertical parameter $t \in [0, 1]$ captures both the effect of breaking C_6 and the effect of the band inversion. The initial ($t = 0$) and final ($t = 1$) phases are C_6 symmetric. Then, for $0 < t$, C_6 symmetry is broken and is followed by the band inversion. This happens through a braiding of the adjacent nodes around the Γ point. Then for $t \rightarrow 1$, the C_6 symmetry is restored. We conclude that, on top of confirming that the band inversion is mediated by a braiding process, this also confirms that the degeneracy at Γ for the C_{6v} symmetric system results from a superposition of two simple nodes which are faithfully counted by the integer patch Euler classes $|\chi_{10}^{(t=0)}[\Gamma]| = |\chi_{11}^{(t=1)}[\Gamma]| = 1$.

7) Some of the authors have written a related paper Ref. 15 and uploaded it in *arXiv*, which also deals with non-Abelian topological physics of phonon. The authors should clarify the novelty and importance of the present work by comparing with Ref. 15.

Answer: We thank the reviewer for raising this point. We note that Ref. 15 (Ref. 3 here) deals with the implementation of a specific three-band model in an *acoustic* meta-material. Here we propose a *real* material candidate that hosts this physics in its intrinsic phonon spectrum, opening the doors for practical applications. [See also our reply to question 1) that addresses the relevance of achieving a controlled band inversion in a real material together with its experimental signature through the Raman spectrum.] Most importantly, this phonon spectrum exhibits, for the first time, braiding processes involving more than three bands. We have now emphasized this in the manuscript, as follows,

We identify a monolayer silicate as a candidate material, and show that an electric field and epitaxial strain can be used to induce band inversions which are accompanied by the transfer of nodes between adjacent gaps, thereby transferring non-Abelian frame charges and non-trivial patch Euler class. This is achieved through the braiding of band nodes, resulting in a multi-gap topological phase. Different from the braiding in the metamaterial,¹⁵ we can realize many-band (more than three) braiding processes in phonons of monolayer silicate.

8) The authors should clarify the relation between the quaternion charge and the Euler number. Some papers discuss that the Euler number is not sufficient to characterize the non-Abelian topology. However, this paper gives an impression that the Euler number is enough.

Answer: We thank the reviewer for pointing this out. We find it unfortunate if some papers give this wrong impression. We would have been interested to know which papers the referee had in mind. On our side, this is no room for any ambiguity. The intrinsic relation between the patch Euler class and the quaternion charges in a three-band model has been rigorously proven in Ref. 2, and it is not necessary to repeat the whole argument here. Therefore the aim of our work goes beyond that specific question. On the one hand, we want to reveal new structures that are consequences of the non-Abelian topology and which can be observed in real material experiments. On the other hand, we do bring forward the theory itself: First, by establishing a systematic methodology to reconstruct the global topological configuration of the non-Abelian nodal phases in the context of arbitrarily many bands, the situation we have to deal with in real materials, and this merely from the computation of patch Euler classes [we also refer to our reply to question 7)]. Second, by providing a new algorithm that corroborates this approach through the direct computation of non-Abelian charges for partial frames, as addressed in our reply to question 2).

9) The patch Euler number is not clearly explained. For example, the authors should explicitly show the region of the patch. The authors should show the integral and the integrand of the Euler number.

Answer: We thank the reviewer for pointing this out. We have taken the opportunity of this revised version to make a better distinction between the patch Euler class evaluated for a single node, and the patch Euler class of a pair of nodes. In the former case, the patch is simply defined as a small square that only covers the node. In the latter case, we have to cover two nodes that are separated in the Brillouin zone. The patch then takes the shape of a rectangle that covers both nodes. In the new version, we make more clear that this is the unique data required to build the global non-Abelian configuration (as long as, at the given snapshot of the phase transition, there are no N -fold degenerated band crossings, in which case, our partial-frame algorithm is needed). Let us emphasize that the region and area of the patch do not affect the value of the patch Euler class as long as the patch contains the same nodes. This is a consequence of the quantization of the patch Euler class as derived in Ref.2.

Besides the above clarifications that now enrich the new version, we have also better emphasized the logic of our strategy, namely the *puzzle* approach. The figures in the updated method section serve as a direct illustration of the two steps on which the strategy is based. First, the computation of the patch Euler classes for all single band crossings, and for all pairs of band crossings. Then, for a chosen initial patch of a pair of nodes, we assign signed frame charges to the nodes and we fix an adjacent Dirac string crossing or avoiding the patch. In order to better introduce this strategy, we have extended the main text as follows,

The topological configurations can be determined by the numerical calculation of the patch Euler class for every single band crossing and for every pair of band crossings of the same gap. We then show how the original data of the computed patch Euler classes can be combined with the assignment of Dirac

strings¹¹ to build the non-Abelian topological configuration of the nodes over the whole Brillouin zone (this is the puzzle approach detailed in Methods). The results are then corroborated through the direct computation of the non-Abelian charge of nodes located in connected multi-band subspaces. The later requires the use of partial-frames (of the connected band subspaces) for which we present an algorithm in Methods.

Let us illustrate this strategy with the example of group I composed of three connected bands. Focusing on gap Δ_{10} in 7% strained silicate under an electric field of 1.0 eV/\AA , we compute the patch Euler class^{11,13,67,68} from the numerically calculated phonon eigenvectors $|u_{10}\rangle$ and $|u_{11}\rangle$, of \mathcal{B}_{10} and \mathcal{B}_{11} respectively (see Methods), for every single band crossing and for every pair of band crossings within the gap. In Fig. 4(a), where we use the convention that the symbols of circle and triangle correspond to nodes in gap Δ_{10} and Δ_{11} respectively, we draw the patches that contain a pair of band crossings located at distinct regions of the Brillouin zone, with the calculated Euler classes indicated for all the patches. The Euler class of the patches containing a single band crossing are indicated by the size of the symbols, i.e. the small circles indicate $|\chi_{10}| = 0.5$ (e.g. the nodes at K) and the large circles with dashed boundary indicate $|\chi_{10}| = 1$ (e.g. the node at Γ), and we use the convention that the fullness and openness of the symbols indicate the signs $+$ and $-$ respectively. We note the correspondence between the dispersion around the nodes and their Euler class, i.e. the nodes with an Euler class of 0.5 exhibit a linear dispersion, while the nodes with an Euler class of ± 1 appear to be quadratic (see also the section on the Euler class-dispersion relation below). We verify this by direct computation (see Methods on the non-Abelian charges of partial-frames) showing that each Euler class of ± 0.5 corresponds to a three-band partial-frame charge $q_{10} = \pm i \in \mathbb{Q}$, while the Euler class of ± 1 corresponds to a three-band partial frame charge of $-1 \in \mathbb{Q}$.

The quadratic nodes can be viewed as two linear nodes merged together. This is explicitly verified under the breaking of C_6 where the double node at Γ splits into two simple nodes (see the braid trajectory under broken C_6 symmetry in Methods). Thus, for every band crossing and for every pair of band crossings within the gap, the patch Euler class takes value in the integers—in which case it indicates an even number of stable nodes, or in the half-integers—in which case it indicates an odd number of stable nodes on the patch, as shown in Fig. 4(a). Since the rotation of a patch by a symmetry of the point group C_6 , leads to an equal Euler class, we only need to consider the patches over the irreducible Brillouin zone. Importantly, the sign of the Euler class on each patch taken separately is gauge dependent, and similarly for the sign of the frame charges q_{10} (see Methods). This motivates the puzzle approach which focuses on the relative stability of the nodes that are gauge invariant.

As a next step, we choose one initial patch containing a pair of band crossings, starting from patch 1 in Fig. 4(a) for which we obtain an Euler class of 0, and we assign a signed non-Abelian charge to each crossing. With an Euler class of 0, we can assign opposite charges to the pair of yellow nodes in patch 1 with no adjacent Dirac string crossing the patch. (Alternatively, we could assign the same charge to these two nodes and separate them by an adjacent Dirac string crossing the patch, see Methods on the gauge freedoms.) The remaining yellow nodes, and the Dirac strings connecting them, are then fixed by the C_6 symmetry. For patch 2, $\chi_{10} = 0.5$ is compatible with the quadratic node (blue) of Euler class $\chi_{10}[\Gamma] = +1$ at Γ and a linear node (yellow) of Euler class $\chi_{10}[\Gamma - K] = -0.5$. For patch 3, $\chi_{10} = 0$, and we can assign opposite frame charges to the yellow node and the dark red node. Similarly, we then assign signed frame charges to all the other nodes, as well as Dirac strings that connect them. While there are relative gauge freedoms in doing so (see the Method section), once the charges of the initial patch is fixed, the charges for all neighbouring patches become fixed by consistency, like completing a puzzle. By repeating this process for all the patches, we get the complete topological configuration, as summarised in Fig. 4(a). We also calculate the 3D band structures in Fig. 4(b) to demonstrate the quadratic node in Δ_{10} at Γ , as well as the

Figure 4. (a) Patches in the Brillouin zone and their corresponding Euler class for 7% strained silicate under an electric field of 1.0 eV/\AA . The Euler class χ_{10} is calculated from the phonon eigenvectors of \mathcal{B}_{10} and \mathcal{B}_{11} in the patches. The resultant patch Euler class in the yellow (patch 1), green (patch 2) and black (patch 3) areas are 0, 0.5 and 0, respectively. **The large blue circle with dashed boundary at Γ is a quadratic node with the patch Euler class of 1.** (b) 3D band structures in the grey areas of (a), with two linear nodes – one (cyan) in Δ_{11} along Γ -M and one (dark red) in Δ_{10} at K, as well as a quadratic node (blue) in Δ_{10} at Γ . These dispersions root in the stability of Euler class, meaning that a stable double node produces a quadratic dispersion. (c) Patches in the Brillouin zone to calculate the Euler class for 7% strained silicate under an electric field of 1.8 eV/\AA . The patch Euler class in the blue (patch 1) and purple (patch 2) patches are both 0. **The large blue circle with dashed boundary at Γ and the large dark red circles with dashed boundary at K are quadratic nodes with the patch Euler class of 1.** (d) 3D band structures in the grey areas of (c), with one linear node (cyan) in Δ_{11} along Γ -M and two quadratic nodes in Δ_{10} at Γ (blue) and K (dark red).

linear nodes in Δ_{10} at K and in Δ_{11} along Γ -M, which relates to their frame charges. We remark that when a node in gap Δ_n crosses a Dirac string connecting nodes residing in neighbouring gap Δ_{n-1} or Δ_{n+1} , the sign of the charge in Δ_n changes. It should also be noticed that the configurations of the Dirac string is not unique, since moving the Dirac string flips the gauge sign of the eigenvectors locally. Nevertheless, the requirement of consistency in the assignment of the signed frame charges together with the location of the Dirac strings, eventually guarantees the full indication of the relative stability of the nodes that is gauge independent.

A similar procedure can be applied to study the topological configurations for 7% strained silicate under an electric field of 1.8 eV/\AA . As shown in Fig. 4(c), patch 1 in Δ_{10} contains a dark red node at K, a quadratic blue node at Γ , and a Dirac string connecting the cyan nodes in Δ_{11} . With an Euler class $\chi_{10} = 0$, the dark red node at K must carry the same frame charge with the blue node at Γ . In addition, patch 2 has an Euler class of 0, indicating that the neighbouring dark red nodes carry opposite frame charges. For the cyan nodes in Δ_{11} , there is no nearby Dirac string in Δ_{10} during the whole braiding process, and their frame charges remain the same.

We finally respectfully point out that the patch Euler class has been explicitly introduced in the section *Patch Euler class* of Methods, with the definition of its two components: (i) the surface integral with its integrand (i.e. the Euler form), and (ii) the loop integral with its integrand (i.e. the Pfaffian connection).

10) Is it possible to observe the Dirac string experimentally in the present system?

Answer: We thank the reviewer for pointing this out. As far as the bulk topology is concerned, the Dirac string is a gauge quantity that cannot be physically observed. In the Methods section, we explain that the configurations of the Dirac strings can be changed without any effect on the physics (i.e. the relative stability of nodes) as it simply accounts for local change of the gauge sign of the Bloch eigenvectors. The only requirement is that a few rules must be satisfied, such as the sign flip of a frame charge whenever it is crossed by an adjacent Dirac string. What it does is to create, together with the signed frame charges, an effective picture where the gauge is fixed globally and consistently. Such a picture has the merit to make the physics of the system explicit on top of being predictive, i.e. the relative stability between distinct nodes can be directly read from the picture and the topological phase after a band inversion can be predicted from the picture without the need to repeat the computation of all the charges. We here refer to our reply to question 9) with the new parts in the main text where we present more clearly our two-step “game”: first a puzzle, followed by a braiding, which both allow an interlude with the change of the picture allowed by the gauge freedoms.

We have clarified this in the main text, as follows,

It should be noted that the Dirac string is a gauge object and hence cannot be physically observed. However, it is necessary for the definition of a global topological configuration of nodes, i.e. the consistent assignment of signed frame charges over the whole Brillouin zone. Such a picture has the merit of capturing the exhaustive topological structure of the phase, i.e. the (gauge invariant) relative stability of any pair of nodes is readily given by their assigned frame charges. Furthermore, any phase transition induced by the moving of the nodes over the Brillouin zone can be predicted from this global picture.

Concerning the edge physics, the Zak phases for a given edge termination can be readily be determined from the Dirac strings. We have clarified this in the main text, as follows,

We can actually predict the Zak phase for the given edge termination from the the bulk topology. The Zak phase at fixed gap is given (modulo 2π) by the parity of the number of Dirac strings that are crossed by the straight path of integration that is perpendicular to the edge axis, i.e. for the (100) edge these are the paths $l_{k_{\parallel}} \in \{k_{\perp}(b_1 - b_2) + k_{\parallel}(b_1/2 + b_2/2) | k_{\perp} \in [0, 1]\}$ at a fixed $k_{\parallel} \in [0, 1]$, with k_{\parallel} the coordinate of the horizontal axis in Fig. 1(c) and Fig. 2(c).¹⁵

11) PT or C_2T symmetry is necessary for the quaternion charge. However, it is not clear which symmetry exists in the present system. The authors should clarify why such a symmetry exists in silicate.

Answer: We thank the reviewer for pointing out this. The system has C_{6v} point group and hence in particular C_2 symmetry. Together with the naturally present time reversal symmetry, this renders C_2T in particular. We have clarified this in the main text, as follows,

Because of the $P6mm$ space group, the system has C_2 rotation symmetry. In addition, in phonons the time reversal symmetry is automatically satisfied. Therefore, monolayer Si_2O_3 has C_2T symmetry and its band nodes at neighbouring gaps can host non-Abelian charges.

12) Is the system robust as long as PT or C_2T symmetry is preserved? The authors should discuss the

effect of impurities and randomness in order to clarify the topological protection. Without it, one cannot determine whether the system is truly topologically protected or accidentally protected.

Answer: We thank the reviewer for pointing this out. On the one hand, we have checked all the irreps that correspond to the C_{6v} symmetry. In the context of solid state materials, it is well established that all the symmetry properties of a system are preserved if the crystal symmetries are preserved at least on average. In other words, random impurities tend to cancel on average. This is the reason why many material properties predicted from first-principle calculations (such as the material stability or its band structure), based on the space group symmetries of perfect crystals, are so universally agreeing with experiments. So much so, that often the space group of the crystal structure of a material can be determined from its powder. (We discard here strongly correlated materials, which are notoriously hard to model.) Going along the same line of logic, the experimental study of Raman scattering is well known to rely heavily on the detailed knowledge of the representation theory of the materials space groups.

On the other hand, our new section in methods on the braid trajectories under the breaking of C_6 symmetry crucially shows that the additional crystalline symmetries are not essential for the manifestations of the non-Abelian topology addressed in this work. The only remaining question thus concerns the robustness of the C_2T symmetry. In that regard, C_2 symmetry and time reversal symmetry are symmetries that are particularly hard to break. Breaking C_2 usually requires to enlarge the unit cell (due to a structural phase transition, or due to doping). This is illustrated by our three-band tight-binding model where we break C_6 symmetry without enlarging the unit cell by imposing distinct onsite energies at the three atomic sites located on the same Wyckoff position. In other words, we artificially set a distinct onsite energy to identical Si atoms which cannot happen in the real material (identical atomic species located at a common Wyckoff position must be symmetric). Addressing now time reversal symmetry, it can be broken either through magnetic impurities, but this requires to change the chemical composition of the material (structural crystal defects for instance are usually not magnetic), or through a spontaneous long-range magnetic order which can be discarded here. We finally note that we do not address the effect of an external magnetic field on the non-Abelian topology. This is a very interesting question but it goes beyond our scope, as phonons are charge neutral, spinless bosonic excitations.

Since this question of stability is generic in solid state physics and that silicates do not constitute an exception, we have kept the discussion of the stability to its minimum, with a discussion of the robustness of non-Abelian charges, as follows,

Although there might be some defects in monolayer silicate due to the imperfection of the growth processes,^{54,59,66} the impurities cannot destroy the non-Abelian charges as long as the C_2T symmetry is preserved.

13) In other papers on non-Abelian topological charges, they use the spin Berry-Wilczek-Zee phase instead of the patch Euler number. The authors should discuss the relevance between them.

Answer: We thank the reviewer for pointing this out. As we have explained in our reply to questions 2) and 5), the Euler class provides the finest topological characterization of the band structure topology of non-Abelian nodal phases. Let us repeat the $q \in \{\pm 1\} = \mathbb{Z}_2$ frame charge characterization of the relative stability of nodes, as compared to the $2\chi \in \mathbb{N}$ characterization of an arbitrary number of stable nodes. Then, when a phase contains $N > 2$ -fold degenerated band crossings, we can supplement the patch Euler class with the computation of the partial frame charges, for which we provide a new algorithm in methods.

Reviewer #2 (Remarks to the Author):

In the manuscript, the authors propose that multi-gap topologies and related node conversion processes can be observed in the phonon spectra of monolayer silicates by applying an electric field and epitaxial strain. The authors made significant effort to describe the complicated node conversion processes in terms of creation and annihilation of topological nodes carrying non-abelian frame charges, and provided an elegant theory for the evolution of phonon band structure under strain and electric field. It is also proposed that a part of the node conversion processes can be monitored by observing the Raman spectra.

I think that the theoretical observation of the node conversion associated with the non-abelian frame charge and multi-gap band topology in phonon spectra is interesting, and the theoretical analysis is rigorous and reliable. I also think that the manuscript appeared in a timely manner as there is a growing interest in the non-abelian topological charges of nodes in recent studies.

However, I also have several concerns about the novelty of this paper, especially because (i) feasibility of the proposed node conversion in real material is not clear, (ii) there is no thorough analysis of the edge states, (iii) the manuscript is still quite technical and theoretical description is complicated. Below I give several questions and comments, which should be properly addressed by the authors, before any decision can be made about the manuscript.

Answer: We would like to thank the reviewers for their work in reviewing our manuscript and for their detailed comments, which help us to improve our work.

(1) In the section about phonon band structure, the low energy phonon bands, from band 1 to 9, are not considered. What happens in these phonon bands when electric field and strain are applied?

Answer: We thank the reviewer for raising this point, and we agree with the reviewer that the low frequency phonons were not discussed in detail. We did not discuss these phonons because they are less sensitive to both the electric field and strain. We have mentioned this in the main text and added Fig. 5 to the Supplementary information, as follows,

We only focus on the three groups of Kagome bands because they are more sensitive to the electric field (see Supplementary information for the full phonon spectra of 5% and 7% strained monolayer silicates at different electric fields).

(2) The stability of the lattice structure and the symmetry of monolayer silicate under strain and electric field should be more carefully analyzed at the level of first-principles calculations, including the low energy phonon bands. Does the material maintain the same structure and symmetry under 7% strain and 2.0 eV/Å electric field, or even under higher strain and electric field? What supports the stability? What is the stability limit?

Answer: We thank the reviewer for raising this point. We have examined the dynamical stability of monolayer silicate by investigating the low-frequency phonon modes, and found no imaginary modes under the studied conditions of strain and electric field, as shown in Fig. 5 and Fig. 6. We have also performed geometry optimizations starting from a rotated Kagome lattice, similar to Ref. 54, and found that structural relaxation at different electric fields always removes the rotation distortion. We have discussed the stability of monolayer silicate and the robustness of non-Abelian charges in the main text, and added Fig. 6 to the Supplementary information, as follows,

We calculate the strain-dependent (from -2% to 8%) phonon dispersion under electric fields from -0.9 to 2.0 eV/Å. No imaginary modes have been observed even for 7% strained Si₂O₃ under an electric

Figure 5. Phonon dispersion of (a) 5% and (b) 7% strained monolayer Si_2O_3 under different electric fields.

field of 2.0 eV/\AA , indicating the dynamical stability of our system (for details, see the Supplementary information). In addition, we distort the crystal structures by creating a rotated Kagome lattice similar to Ref. 4, and structural relaxation at different electric fields always removes the rotation distortion. Experimentally it has been found that both monolayer silicate and bilayer silica can be grown on various substrates with biaxial strains varying from -5.6% to 5.7% .^{5–10} In addition, theoretical calculations have shown that defect-free 2D silica can be deformed up to a maximum strain of 10.4% , while for defective samples no abrupt material failure is observed at strains around $7.8\text{--}8.1\%$.¹¹ Regarding the electric field, by applying bias voltage between the tip of the scanning tunneling microscopy (STM) and the sample, a maximum electric field of 1.7 eV/\AA can be obtained.¹² Therefore, both the strain and the electric field used in our calculations are experimentally feasible.

...

Because of the $P6mm$ space group, the system has C_2 rotation symmetry. In addition, in phonons the time reversal symmetry is automatically satisfied. Therefore, monolayer Si_2O_3 has C_2T symmetry and its band nodes at neighbouring gaps can host non-Abelian charges. Although there might be some defects in monolayer silicate due to the imperfection of the growth processes,^{4,13,14} the impurities cannot destroy the non-Abelian charges as long as the C_2T symmetry is preserved.

(3) Is it possible to apply 7% strain and 2.0 eV/\AA electric field simultaneously in experiments? The experimentally accessible region of strain and electric field should be clearly mentioned.

Answer: Two-dimensional materials can withstand much larger strain compared to their bulk counterparts. For example, most graphite materials break when the strain is up to 0.1% , while monolayer graphene can withstand 25% elastic strain.¹⁵ For the material family we are interested in, experimentally

Figure 6. Phonon dispersion of 7% strained monolayer Si_2O_3 under an electric field of 2.0 eV/\AA .

it has been found that both monolayer silicate and bilayer silica can be grown on various substrates with biaxial strains varying from -5.6% to 5.7% .^{5–10} In addition, theoretical calculations have shown that defect-free 2D silica can be deformed up to a maximum strain of 10.4% , while for defective samples no abrupt material failure is observed at strains around $7.8\text{--}8.1\%$.¹¹ Regarding the electric field, by applying a bias voltage between the tip of the scanning tunneling microscopy (STM) and the sample, a maximum electric field of 1.7 eV/\AA can be obtained.¹² Therefore, both the strain and the electric field used in our calculations are experimentally feasible. We have clarified these points, as demonstrated in our replies to previous comment.

(4) In the section about “Braiding in group 1”, it is stated that “Applying an electric field increases the frequency of the non-degenerate band B_{10} at Gamma while reducing the frequency of the degenerate bands $B_{11,12}$ ”. What is the physical reason for the change of the energy ordering under electric field? Is it possible to explain it physically by considering the deformation pattern of the relevant phonon modes?

Answer: The B_{10} vibrational mode at Γ corresponds to an out-of-plane displacement of silicon atoms and an opposite-direction displacement of oxygen atoms, and the $B_{11,12}$ mode consists of both the in-plane displacements of silicon atoms and the out-of-plane displacements of oxygen atoms, as shown in Fig. 7. In monolayer Si_2O_3 , Si and O atoms carry out-of-plane Born effective charges of $+1.0$ and -0.5 , respectively. As a result, the out-of-plane electric field increases the out-of-plane distance between Si and O atoms, leading to weakened interatomic interactions and reduced phonon frequencies for $B_{11,12}$ at Γ . However, this out-of-plane electric field can also enhance the opposite out-of-plane motions of two types of atoms with opposite signs of Born effective charge, which hardens the vibrational mode and increases the frequency of B_{10} at Γ .

We have explained the microscopic mechanism of the increased frequency of B_{10} in the main text and added Fig. 7 to the Supplementary information, as follows,

The B_{10} vibrational mode at Γ corresponds to an out-of-plane displacement of silicon atoms and an opposite-direction displacement of oxygen atoms, and the $B_{11,12}$ mode consists of both the in-plane displacements of silicon atoms and the out-of-plane displacements of oxygen atoms (for details, see the vibrational patterns in the Supplementary information). For the B_{10} mode, the two types of atoms carry out-of-plane Born effective charges of $+1.0$ and -0.5 , respectively, and an out-of-plane electric field can enhance the opposite out-of-plane motions of two types of atoms with opposite signs of Born effective charge, which hardens the vibrational mode and increases the frequency of B_{10} at Γ . For the $B_{11,12}$ mode at Γ , the out-of-plane electric field increases the out-of-plane distance between Si and O atoms, leading to weakened interatomic interactions and reduced phonon frequencies. Upon the electric field, the frequency of the lower B_{10} increases while the frequency of the higher $B_{11,12}$ decreases, and a phonon

band inversion takes place at an electric field of 0.7 eV/Å. As a result, new nodes are formed by the inverted bands with different irreps.

Figure 7. Phonon vibrational mode of (a) \mathcal{B}_{10} and (b) $\mathcal{B}_{11,12}$ at Γ .

(5) Although the authors claimed that braiding of nodes can be controlled, what is actually happening in the system is complicated conversion processes of many nodes simultaneously. In the usual description of braiding in real space, one can imagine two point-like objects whose positions can be interchanged. Can it be possible to perform such a braiding of individual node in the proposed phonon system, instead of the simultaneous conversion of many nodes? If such a process is impossible, I am wondering how useful the momentum space braiding is, compared to the relevant real space counterpart.

Answer: We thank the referee for raising this very interesting point. We first address the question of the braiding in momentum space. For clarity, we are going to focus on the braiding in group 1, but similar conclusions hold in general. It is important to distinguish here between the control of a single node and the control of a single braiding, the latter of which in our case can be directly linked to a specific band inversion. Indeed, the point group C_{6v} of the monolayer Si_2O_3 imposes that groups of nodes be created or annihilated together upon a band inversion. However, the picture simplifies by only considering the irreducible Brillouin zone or fundamental domain. For this reason, it is not necessarily true that there is more information stored through a band inversion-braiding in this case, as compared to a system with only C_2T symmetry. Concretely, in the new version of the work we show the braid trajectories in group 1 upon the band inversion at Γ and under the breaking of C_6 symmetry, using an effective three-band tight-binding model (see our reply to question 6 of Referee 1, with Fig. 3 and the related new section in Methods). There, it is revealed that the band inversion (transferring a double node, i.e. a pair of stable nodes, from gap 11 to gap 10) happens through a single braiding in the vicinity of Γ . As more clearly explained in the new version, the other features—that the braid trajectories in the silicates are collapsed on the high-symmetry points, and that the band inversion is accompanied by an emission of 12 simple nodes [see Fig. 2(b) of the main text)]—are effects of the crystal point symmetries. These additional features disappear by relaxing the point symmetries while keeping C_2T . In that regard we note that while the C_6 symmetry can be broken easily through strain, breaking C_2 symmetry while preserving C_2T is harder within real materials.

We do see an advantage of the band inversion-braiding process in a system with symmetry though. As the creation/annihilation and transfer of nodes is controlled by the band inversions, by symmetry, only very specific terms (say in the context of an effective tight-binding model) are involved in the process, whereas in a system without point symmetries the parameter space of processes (of tight-binding terms) is much larger, i.e. more symmetries means more stringent selection rules. Therefore, there is no guarantee that a braiding process could be controlled simply through an isomorphic strain combined with an electric field

in a system with fewer symmetries. The advantage of a symmetric system is even stronger considering that the band inversion-braiding can be monitored through the Raman spectrum at Γ .

Finally, we briefly address the question of the comparison between braiding in real space and braiding in momentum space. The main difference lies in the fact that the momentum space braiding happens at the level of the bulk topology of the system, while the braiding in real space is conditioned by the realization of boundary states. In the case of real space braiding, the bulk-boundary correspondence constitutes an extra layer of complexity that must be controlled. Considering for instance the braiding of Majoranas that has attracted wide interest, their experimental observation has proven challenging. On the contrary, the observation of momentum space band inversion-braiding in real materials may not be very far ahead. As shown in our work, such an achievement was probably impeded by a lack of theory rather than from experimental limitations.

(6) To describe the linear and quadratic nodes, the authors used the convention that linear nodes are represented by small symbols and quadratic nodes by large symbols. But it seems to be better to use another convention for clarity.

Answer: We thank the reviewer for pointing out this. We have added a dashed boundary for the quadratic nodes in Fig. 8(b), and clarified which nodes are quadratic in the caption. For the quadratic nodes in Fig. 9(b), they are all at the Γ point, so we can just clarify this in the caption.

Figure 8. The topological configurations with nodes in different gaps (different symbols) and Dirac string configurations are shown in (b). **The large blue triangle (at 0.0 eV/\AA) and the large blue circles (at $0.7 - 1.8 \text{ eV/\AA}$) with dashed boundary at Γ , as well as the large dark red circles with dashed boundary at K (at 1.8 eV/\AA), are quadratic nodes.**

Figure 9. The panels in (b) show the nodes and their topological charges. As before, different symbols characterise topological charges in different gaps, whereas open versus closed symbols indicate the sign of the charge. Dirac strings are drawn as lines. **The large blue triangles and the large red squares at Γ are quadratic nodes.**

(7) In the section about edge states, the authors mentioned that “we take a spectator view by directly visualizing the edge states but without addressing the fundamental mechanisms behind them”. But the proper Zak phase analysis for the proposed phonon band structure should be performed. What is the origin of the edge arc connecting two adjacent nodes at the neighboring Gamma points on the (100) edge? Proper explanation for the other edge spectra also should be provided.

Answer: We thank the reviewer for pointing this out. We have performed the Zak phase calculations, and found that for the three-band braiding the Zak phase for each gap can describe the edge states of the three bands effectively whereas for the four-band braiding the Zak phase fails to provide a consistent picture. We have clarified these points, as follows,

Edge states

We also study the evolution of topological edge states with varying electric field by calculating the surface local density of states (LDOS) from the imaginary part of the surface Green’s function.⁷¹ Figure 10(c) shows the edge states of 7% strained silicate. On the (100) edge (i.e. the zigzag direction for the Si atoms), there is an edge arc connecting two adjacent nodes at the neighbouring Γ points. Upon increasing electric field, new band crossing points appear around Γ . At 1.8 eV/Å, there are two clear projections of the new Dirac cones on the (100) edge around 17.36 THz. We calculate the Zak phase γ (i.e. the Berry phase along a non-contractible path of the Brillouin zone which is here quantised to $\{0, \pi\}$ by the C_2T symmetry) in gap Δ_{10} and Δ_{11} at 0.0 and 1.8 eV/Å. As shown in Fig. 10(c), a Zak phase of zero corresponds to the emergence of the edge states, while $\gamma = \pi$ leads to vanishing edge states. This indicates that the band Wannier states (the Wannier states are the Fourier transform of the Bloch eigenstates) have their center (the “band centers”) shifted from the center of the atomic Wannier states (the “atomic centers”), leading to a charge anomaly and, following, the localisation of an edge state. Furthermore, the fact that the edge states appear when the Zak phase is zero simply means that the band centers occupy the center of the unit cell, while the atomic centers lie on the boundary of the unit cell.^{15,72} Indeed, the \mathbb{Z}_2 quantised Berry phase is a good quantum number in 1D, which indicates the band center (i.e. its Wyckoff position).⁷³

We can actually predict the Zak phase for the given edge termination from the the bulk topology. The Zak phase at a fixed gap is given (modulo 2π) by the parity of the number of Dirac strings that are crossed by the straight path of integration that is perpendicular to the edge axis, i.e. for the (100) edge these are the paths $l_{k_{\parallel}} \in \{k_{\perp}(b_1 - b_2) + k_{\parallel}(b_1/2 + b_2/2) | k_{\perp} \in [0, 1]\}$ at a fixed $k_{\parallel} \in [0, 1]$, with k_{\parallel} the coordinate of the horizontal axis in Fig. 10(c) and Fig. 11(c).¹⁵

Figure 10. The corresponding phonon edge states on the (100) edge are shown in (c), with the $\{0, \pi\}$ -quantised Zak phases $\{\gamma_n\}_{n=10,11}$ for gaps $\{\Delta_n\}_{n=10,11}$ indicated by the arrows.

Figure 11(c) shows the topological edge states of 5% strained silicate on the (100) edge. Besides the edge arc that connects two adjacent nodes at the neighbouring Γ points around 26.45 THz, there is also a new edge arc connecting two adjacent nodes around 25.90 THz located at neighbouring K points with 2D irrep K_3 . Under an electric field of 0.9 eV/\AA , extra edge arcs emerge near $\bar{\Gamma}$ at 25.97 THz. The arc connecting K_3 moves downwards upon increasing electric field and is robust. For the four-band braiding processes, the Zak phase argument fails, as the Zak phases γ_{14-17} in Fig. 11(c) do not show a consistent behaviour. We stress that, apart from effective Zak phase diagnoses,^{15,74} the full multi-gap bulk-boundary correspondence remains an open question. Hence, we take here a “spectator” view by directly visualising the edge states but without addressing the fundamental mechanisms behind them.

Figure 11. The corresponding phonon edge states on the (100) edge are shown in (c), with the $\{0, \pi\}$ -quantised Zak phases $\{\gamma_n\}_{n=14,\dots,17}$ for the gaps $\{\Delta_n\}_{n=14,\dots,17}$ indicated by the arrows.

(8) The Raman spectrum can provide only a part of the information about the node conversion. This point should be clearly mentioned in the manuscript.

Answer: We agree with the reviewer that the Raman spectra merely provides information on the band inversions at Γ . However, through the knowledge of the irreps at Γ and of the non-Abelian nature of the system, we directly deduce the creation/annihilation of nodes and their transfer between adjacent gaps. Therefore, the measurement of the band inversions provides the monitoring of the corresponding symmetry constrained braiding processes. See in particular our reply to question 6) of Referee 1, and the new section of Methods *Explicit braiding patterns via C_6 symmetry breaking*, where we prove that the band inversion at Γ , i.e. the transfer of a double node (a stable pair of nodes) involves a braiding. See also our reply to question 1) of Referee 1.

Additionally, we demonstrate, for the first time, that Raman spectra can be used to track indirectly the braiding processes controlled by experimentally feasible external parameters such as strain and electric field. Therefore, we have clarified that the Raman spectra can only detect the band inversion processes and that the braiding processes are accompanied by the band inversion, as follows,

(Abstract)

Finally, we propose that *the band inversion processes at the Γ point can be tracked by following the evolution of the Raman spectrum, providing a clear signature for the experimental verification of the band inversion accompanied by the braiding process.*

...

(Introduction)

We further show that the evolution of Raman peaks can be used to track the band inversions and *the accompanying braiding process*, providing a clear experimental signature to identify multi-gap topology in real materials.

...

(Experimental signature: Raman spectra)

We finally propose that the *band inversion* processes described above can be directly observed experimentally by following the evolution of the Raman spectrum of the material.

...

These calculations show that Raman spectroscopy can be a promising tool for characterizing *the band inversion processes and the accompanying non-Abelian braiding of phonons* in 2D materials such as monolayer silicate.

(9) The manuscript is still quite technical and complicated, which disturb the readability of the manuscript.

Answer: We thank the reviewer for pointing out this, and we agree with the reviewer that some part were technical. We have improved this by relegating all technical details of the theory to the Method section. Moreover, we have reformulated this section extensively to better explain the concepts. Finally, we note that we have included a new theoretical tool which allows us to corroborate the global topological structure obtained from the patch Euler class. Indeed, via the use of partial-frame we can directly compute the non-Abelian frame charges of nodes within any M -band subspace with $M \leq N$, where N is the total number of bands. See also our response to question 2) of Referee 1. We thank the referee for giving us the opportunity to drastically improve our manuscript.

Reviewer #3 (Remarks to the Author):

In this work Peng *et al* study the non-Abelian topological charges in a layered silicates system, via straining and/or an applied DC bias. They theoretically calculate the band structure of layered silicates, which consists of three groups of bands. Focusing on the three-band in the lowest group, the authors show that straining can reduce the band energy difference close to the gamma point. By further applying a voltage, the band crossing can be induced, leading to creation of additional band nodes between the three bands. Varying the strain and/or voltage drives the motion of the band nodes in the Brillouin zone. The induced braiding behaviour between nodes of different bands are studied in theory. The author further present the simulation results of the Raman spectra at different applied strains and voltages, and claim that the braiding behaviour can be directly observed in experiment via the evolution of Raman spectra.

The work proposes a realistic system for observing the interesting braiding band structure for phonons. The physics is interesting but the presentation of the paper is far from being accessible to readers. Before I can make a recommendation, the authors need to address the following points.

Answer: We would like to thank the reviewer for their work in reviewing our manuscript and for their comments, which provide us opportunities to improve our manuscript.

1. How realistic is the proposed strain level, e.g. 5% and 7%?

Answer: We thank the reviewer for pointing this out. We have examined the dynamical stability of 5% and 7% strained monolayer silicate by investigating the low-frequency phonon modes, and found no imaginary modes, as shown in Fig. 5 and Fig. 6. In addition, two-dimensional materials can withstand much larger strain compared to their bulk counterparts. For example, most graphite materials break when the strain is up to 0.1%, while monolayer graphene can withstand 25% elastic strain.¹⁵ For the material family we are interested in, experimentally it has been found that both monolayer silicate and bilayer silica can be grown on various substrates with biaxial strains varying from -5.6% to 5.7%.⁵⁻¹⁰ In addition, theoretical calculations have shown that defect-free 2D silica can be deformed up to a maximum strain of 10.4%, while for defective samples no abrupt material failure is observed at strains around 7.8-8.1%.¹¹ We have clarified these points, as follows,

No imaginary modes have been observed even for 7% strained Si₂O₃ under an electric field of 2.0 eV/Å, indicating the dynamical stability of our system (for details, see the Supplementary information).

...

Experimentally it has been found that both monolayer silicate and bilayer silica can be grown on various substrates with biaxial strains varying from -5.6% to 5.7%.⁵⁻¹⁰ In addition, theoretical calculations have shown that defect-free 2D silica can be deformed up to a maximum strain of 10.4%, while for defective samples no abrupt material failure is observed at strains around 7.8-8.1%.¹¹ Regarding the electric field, by applying bias voltage between the tip of the scanning tunneling microscopy (STM) and the sample, a maximum electric field of 1.7 eV/Å can be obtained.¹² Therefore, both the strain and the electric field used in our calculations are experimentally feasible.

2. The presentation is very unclear, and this makes it very difficult to follow the paper. For example, Fig. 2b was not explained in detail. How is the Dirac string configuration obtained? Are the configurations of the Dirac string plotted in Fig. 2b unique?

Answer: We thank the reviewer for pointing this out, and we agree with the reviewer that in our original manuscript, the procedure to obtain the topological configurations was not discussed in detail.

The Dirac string is obtained based on the relative stability of different pairs of nodes calculated from the patch Euler class. The Dirac string is a gauge object and as a result, the same patch can have different Dirac string configurations. For example, an Euler class $\chi_{10} = 0$ indicates that a pair of nodes in Δ_{10} can either carry opposite frame charges, or have the same frame charge with a Dirac string in Δ_{11} crossing the patch. Despite the different Dirac string configurations, the relative stability of the nodes is the same, because the node must flip its sign of the frame charge when crossing a Dirac string in the neighbouring gap. We have explained in detail of how we obtain the topological configurations in Fig. 12, and how we fix the corresponding Dirac string configurations, as follows,

The topological configurations can be determined by the numerical calculation of the patch Euler class for every single band crossing and for every pair of band crossings of the same gap. We then show how the original data of the computed patch Euler classes can be combined with the assignment of Dirac strings¹¹ to build the non-Abelian topological configuration of the nodes over the whole Brillouin zone (this is the puzzle approach detailed in Methods). The results are then corroborated through the direct computation of the non-Abelian charge of nodes located in connected multi-band subspaces. The later requires the use of partial-frames (of the connected band subspaces) for which we present an algorithm in Methods.

Let us illustrate this strategy with the example of group 1 composed of three connected bands. Focusing on gap Δ_{10} in 7% strained silicate under an electric field of 1.0 eV/\AA , we compute the patch Euler class^{11,13,67,68} from the numerically calculated phonon eigenvectors $|u_{10}\rangle$ and $|u_{11}\rangle$, of \mathcal{B}_{10} and \mathcal{B}_{11} respectively (see Methods), for every single band crossing and for every pair of band crossings within the gap. In Fig. 12(a), where we use the convention that the symbols of circle and triangle correspond to nodes in gap Δ_{10} and Δ_{11} respectively, we draw the patches that contain a pair of band crossings located at distinct regions of the Brillouin zone, with the calculated Euler classes indicated for all the patches. The Euler class of the patches containing a single band crossing are indicated by the size of the symbols, i.e. the small circles indicate $|\chi_{10}| = 0.5$ (e.g. the nodes at K) and the large circles with dashed boundary indicate $|\chi_{10}| = 1$ (e.g. the node at Γ), and we use the convention that the fullness and openness of the symbols indicate the signs $+$ and $-$ respectively. We note the correspondence between the dispersion around the nodes and their Euler class, i.e. the nodes with an Euler class of 0.5 exhibit a linear dispersion, while the nodes with an Euler class of ± 1 appear to be quadratic (see also the section on the Euler class-dispersion relation below). We verify this by direct computation (see Methods on the non-Abelian charges of partial-frames) showing that each Euler class of ± 0.5 corresponds to a three-band partial-frame charge $q_{10} = \pm i \in \mathbb{Q}$, while the Euler class of ± 1 corresponds to a three-band partial frame charge of $-1 \in \mathbb{Q}$.

The quadratic nodes can be viewed as two linear nodes merged together. This is explicitly verified under the breaking of C_6 where the double node at Γ splits into two simple nodes (see the braid trajectory under broken C_6 symmetry in Methods). Thus, for every band crossing and for every pair of band crossings within the gap, the patch Euler class takes value in the integers—in which case it indicates an even number of stable nodes, or in the half-integers—in which case it indicates an odd number of stable nodes on the patch, as shown in Fig. 12(a). Since the rotation of a patch by a symmetry of the point group C_{6v} leads to an equal Euler class, we only need to consider the patches over the irreducible Brillouin zone. Importantly, the sign of the Euler class on each patch taken separately is gauge dependent, and similarly for the sign of the frame charges q_{10} (see Methods). This motivates the puzzle approach which focuses on the relative stability of the nodes that are gauge invariant.

As a next step, we choose one initial patch containing a pair of band crossings, starting from patch 1 in Fig. 12(a) for which we obtain an Euler class of 0, and we assign a signed non-Abelian charge to each

Figure 12. (a) Patches in the Brillouin zone and their corresponding Euler class for 7% strained silicate under an electric field of 1.0 eV/\AA . The Euler class χ_{10} is calculated from the phonon eigenvectors of \mathcal{B}_{10} and \mathcal{B}_{11} in the patches. The resultant patch Euler class in the yellow (patch 1), green (patch 2) and black (patch 3) areas are 0, 0.5 and 0, respectively. **The large blue circle with dashed boundary at Γ is a quadratic node with the patch Euler class of 1.** (b) 3D band structures in the grey areas of (a), with two linear nodes – one (cyan) in Δ_{11} along Γ - M and one (dark red) in Δ_{10} at K , as well as a quadratic node (blue) in Δ_{10} at Γ . These dispersions root in the stability of Euler class, meaning that a stable double node produces a quadratic dispersion. (c) Patches in the Brillouin zone to calculate the Euler class for 7% strained silicate under an electric field of 1.8 eV/\AA . The patch Euler class in the blue (patch 1) and purple (patch 2) patches are both 0. **The large blue circle with dashed boundary at Γ and the large dark red circles with dashed boundary at K are quadratic nodes with the patch Euler class of 1.** (d) 3D band structures in the grey areas of (c), with one linear node (cyan) in Δ_{11} along Γ - M and two quadratic nodes in Δ_{10} at Γ (blue) and K (dark red).

crossing. With an Euler class of 0, we can assign opposite charges to the pair of yellow nodes in patch 1 with no adjacent Dirac string crossing the patch. (Alternatively, we could assign the same charge to these two nodes and separate them by an adjacent Dirac string crossing the patch, see Methods on the gauge freedoms.) The remaining yellow nodes, and the Dirac strings connecting them, are then fixed by the C_6 symmetry. For patch 2, $\chi_{10} = 0.5$ is compatible with the quadratic node (blue) of Euler class $\chi_{10}[\Gamma] = +1$ at Γ and a linear node (yellow) of Euler class $\chi_{10}[\Gamma - K] = -0.5$. For patch 3, $\chi_{10} = 0$, and we can assign opposite frame charges to the yellow node and the dark red node. Similarly, we then assign signed frame charges to all the other nodes, as well as Dirac strings that connect them. While there are relative gauge freedoms in doing so (see the Method section), once the charges of the initial patch is fixed, the charges for all neighbouring patches become fixed by consistency, like completing a puzzle. By repeating this process for all the patches, we get the complete topological configuration, as summarised in Fig. 12(a). We also calculate the 3D band structures in Fig. 12(b) to demonstrate the quadratic node in Δ_{10} at Γ , as well as the linear nodes in Δ_{10} at K and in Δ_{11} along Γ - M , which relates to their frame charges. We remark that when a node in gap Δ_n crosses a Dirac string connecting nodes residing in neighbouring gap Δ_{n-1} or Δ_{n+1} , the sign of the charge in Δ_n changes. It should also be noticed that the configurations of the

Dirac string is not unique, since moving the Dirac string flips the gauge sign of the eigenvectors locally. Nevertheless, the requirement of consistency in the assignment of the signed frame charges together with the location of the Dirac strings, eventually guarantees the full indication of the relative stability of the nodes that is gauge independent.

A similar procedure can be applied to study the topological configurations for 7% strained silicate under an electric field of 1.8 eV/\AA . As shown in Fig. 12(c), patch 1 in Δ_{10} contains a dark red node at K, a quadratic blue node at Γ , and a Dirac string connecting the cyan nodes in Δ_{11} . With an Euler class $\chi_{10} = 0$, the dark red node at K must carry the same frame charge with the blue node at Γ . In addition, patch 2 has an Euler class of 0, indicating that the neighbouring dark red nodes carry opposite frame charges. For the cyan nodes in Δ_{11} , there is no nearby Dirac string in Δ_{10} during the whole braiding process, and their frame charges remain the same.

3. The Raman signal only traces the shift of the bands at Gamma point, while the key features of braiding in Fig. 2-4 are manifested in the motion of the degeneracy nodes away from the Gamma point. So the claim “the braiding processes described above can be directly observed experimentally by following the evolution of the Raman spectrum of the material” is quite overstated.

Answer: We agree with the reviewer that the Raman spectra merely provides information on the band inversions at Γ . However, through the knowledge of the irreps at Γ and of the non-Abelian nature of the system, we directly deduce the creation/annihilation of nodes and their transfer between adjacent gaps. Therefore, the measurement of the band inversions provides the monitoring of the corresponding symmetry constrained braiding processes. See in particular our reply to question 6) of Referee 1, and the new section of Methods *Explicit braiding patterns via C_6 symmetry breaking*, where we prove that the band inversion at Γ , i.e. the transfer of a double node (a stable pair of nodes) involves a braiding. See also our reply to question 1) of Referee 1.

Additionally, we demonstrate, for the first time, that Raman spectra can be used to track indirectly the braiding processes controlled by experimentally feasible external parameters such as strain and electric field. Therefore, we have clarified that the Raman spectra can only detect the band inversion processes and that the braiding processes are accompanied by the band inversion, as follows,

(Abstract)

*Finally, we propose that **the band inversion processes at the Γ point** can be tracked by following the evolution of the Raman spectrum, providing a clear signature for the experimental verification of **the band inversion accompanied by the braiding process**.*

...

(Introduction)

*We further show that the evolution of Raman peaks can be used to track the band inversions and **the accompanying braiding process**, providing a clear experimental signature to identify multi-gap topology in real materials.*

...

(Experimental signature: Raman spectra)

*We finally propose that the **band inversion** processes described above can be directly observed experimentally by following the evolution of the Raman spectrum of the material.*

...

*These calculations show that Raman spectroscopy can be a promising tool for characterizing **the band inversion processes and the accompanying non-Abelian braiding of phonons** in 2D materials such as monolayer silicate.*

References

1. Wu, Q., Soluyanov, A. A. & Bzdušek, T. Non-abelian band topology in noninteracting metals. *Science* **365**, 1273– (2019).
2. Bouhon, A. *et al.* Non-abelian reciprocal braiding of weyl points and its manifestation in zrte. *Nature Physics* **16**, 1137–1143 (2020).
3. Jiang, B. *et al.* Observation of non-abelian topological semimetals and their phase transitions. *arXiv:2104.13397* (2021).
4. Björkman, T. *et al.* Vibrational properties of a two-dimensional silica kagome lattice. *ACS Nano* **10**, 10929–10935 (2016).
5. Heyde, M., Shaikhutdinov, S. & Freund, H.-J. Two-dimensional silica: Crystalline and vitreous. *Chemical Physics Letters* **550**, 1–7 (2012).
6. Shaikhutdinov, S. & Freund, H.-J. Ultrathin silica films on metals: The long and winding road to understanding the atomic structure. *Adv. Mater.* **25**, 49–67 (2013).
7. Ben Romdhane, F. *et al.* In situ growth of cellular two-dimensional silicon oxide on metal substrates. *ACS Nano* **7**, 5175–5180 (2013).
8. Büchner, C. & Heyde, M. Two-dimensional silica opens new perspectives. *Progress in Surface Science* **92**, 341–374 (2017).
9. Zhou, C. *et al.* Tuning two-dimensional phase formation through epitaxial strain and growth conditions: silica and silicate on nixpd_{1-x}(111) alloy substrates. *Nanoscale* **11**, 21340–21353 (2019).
10. Guo, H. *et al.* Insulating sio₂ under centimeter-scale, single-crystal graphene enables electronic-device fabrication. *Nano Lett.* **20**, 8584–8591 (2020).
11. Bamer, F., Ebrahim, F. & Markert, B. Athermal mechanical analysis of stone-wales defects in two-dimensional silica. *Computational Materials Science* **163**, 301–307 (2019).
12. Kaya, D. The effect of electric field on a fullerene molecule on a metal surface by a nano stm tip. *Physica B: Condensed Matter* **557**, 126–131 (2019).
13. Mathur, S. *et al.* Degenerate epitaxy-driven defects in monolayer silicon oxide on ruthenium. *Phys. Rev. B* **92**, 161410 (2015).
14. Kremer, G. *et al.* Electronic band structure of ultimately thin silicon oxide on ru(0001). *ACS Nano* **13**, 4720–4730 (2019).
15. Novoselov, K. S. *et al.* Electric Field Effect in Atomically Thin Carbon Films. *Science* **306**, 666 (2004).

REVIEWER COMMENTS

Reviewer #1 (Remarks to the Author):

I cannot recommend the publication of this manuscript after reading the responses of the authors due to the following reasons.

I recommend to transfer this manuscript to a more specialized journal.

1)

This paper is too hard to read and understand for general audience as pointed out also by the other referees. This is also the case for the revised version. This greatly degrades the value of this manuscript in the context of the broad interest.

2)

There are already many reports on the non-Abelian braiding of semimetals as cited in this paper. The authors claim that this paper firstly reports the non-Abelian braiding in the phononic spectrum in inorganic materials. However, I do not think that an inorganic material has an advantage comparing with artificial systems such as acoustic or photonic metamaterial because artificial systems are more controllable. In addition, applying 7% stress is not easy for experimentalists. In this context, this paper does not have significant novelty in the context of the current understanding of the non-Abelian topological physics.

3)

As stated by the authors, the Raman spectrum does not show an evidence of the non-Abelian braiding but only shows a band structure modification.

It is not enough as a smoking gun experiment of the non-Abelian braiding. The authors raise an example of ARPES, where only the surface state is observed. However, this is not the case because the emergence of the surface state is a strong evidence of a topological phase. On the other hand, there is no way to confirm the non-Abelian nature of the braiding experimentally. This is not enough for the publication of this manuscript in Nature Communications. The authors should propose a direct method to experimentally observe non-Abelian algebra of the braiding.

4)

The characteristic feature of the edge state for the non-Abelian topological system is still not clear enough. It seems that the edge state only reflects the Zak phase, which is irrelevant to the patched Euler number. How can we detect the non-Abelian nature from the evolution of the edge state?

5)

It is hard to see the braiding structure from Fig.7. In the braiding, the number of the strings should not change. On the other hand, the number of red spots changes. It looks like not a braiding but a merging process of the red spots.

Reviewer #2 (Remarks to the Author):

I thank the authors for their significant efforts to answer my questions and improve the manuscript. I think the authors answered my questions clearly. I recommend the publication of the revised manuscript.

Reviewer #3 (Remarks to the Author):

The authors have made extensive revisions to address all the reviewers' comments, in particular how

the evolution of Raman spectrum is linked to the braiding process. I am happy to recommend the publication of the work to Nature Communication.

Reviewer #1 (Remarks to the Author):

I cannot recommend the publication of this manuscript after reading the responses of the authors due to the following reasons.

I recommend to transfer this manuscript to a more specialized journal.

Answer: We would like to thank the reviewers for their work in reviewing our manuscript and for their detailed comments. Reviewer 1 has not directly responded to our previous reply, so we assume that they are satisfied with our answers to their first round of questions. Here, we further address the new set of comments, and we believe that our work merits publication in *Nature Communications*, a point of view shared by Reviewer 2 and Reviewer 3.

1) This paper is too hard to read and understand for general audience as pointed out also by the other Reviewers. This is also the case for the revised version. This greatly degrades the value of this manuscript in the context of the broad interest.

Answer: We are disappointed that the Reviewer is of the opinion that our manuscript is hard to read. In response to the previous review round, we have done the following to make the manuscript clearer:

1. We have moved all technical details to the Methods section to make the main text clearer.
2. We have presented a new and easily implementable methodology in terms of partial frame charges to calculate non-Abelian properties.
3. We have substantially revised the text to make it more general.

These changes have been recognised by the other Reviewers, who agree with us that the manuscript is now clear and recommend publication. We can therefore only disagree with this comment, and given the lack of detail about what is still considered unclear, we have made no further changes to the text.

2) There are already many reports on the non-Abelian braiding of semimetals as cited in this paper. The authors claim that this paper firstly reports the non-Abelian braiding in the phononic spectrum in in-organic materials. However, I do not think that an in-organic material has an advantage comparing with artificial systems such as acoustic or photonic metamaterial because artificial systems are more controllable. In addition, applying 7% stress is not easy for experimentalists. In this context, this paper does not have significant novelty in the context of the current understanding of the non-Abelian topological physics.

Answer: Here we respectfully, but strongly, disagree with the Reviewer. This comment contains a number of rather different points, so we address them in turn below, dividing the comment into the different parts:

There are already many reports on the non-Abelian braiding of semimetals as cited in this paper.

Answer: The fact that there are already many works on a topic does not mean that a work cannot be novel. This crucially depends on whether the new work advances the field, but we note that the Reviewer has failed to specify how our work fails to advance the field in their view. We instead reiterate the novelty of our work:

- In terms of quasiparticle platforms, we show, for the first time, that phonons are generally a viable platform for non-Abelian physics, and may in fact be the *best* platform for observing these

phenomena in real materials (e.g. the probe of multi-gap physics in fermionic states is more challenging).

- In terms of material realization and experimental observation, we present, for the first time, a real material candidate exhibiting non-Abelian braiding in its phonon spectrum, and additionally describe the evolution of the Raman spectrum as a clear experimental signature.
- In terms of computational methodology, we provide a new partial frame methodology to study non-Abelian charges in *many bands* in a simple manner. This methodology is widely applicable and resolves the shortcomings of earlier methods, and we anticipate that it will play a major role in the study of non-Abelian charges moving forward.

We would also like to add that the increasing number of papers on non-Abelian braiding (most of them appearing this year) reflects the fact that this is a booming field, which in our view only makes our contribution more timely.

The authors claim that this paper firstly reports the non-Abelian braiding in the phononic spectrum in in-organic materials. However, I do not think that an in-organic material has an advantage comparing with artificial systems such as acoustic or photonic metamaterial because artificial systems are more controllable.

Answer: It is not true that metamaterials are generally easier to manipulate than real materials. In fact, this is *definitely* not true in the present context: in metamaterials, each set of parameters requires the construction of a *different* metamaterial, making any practical implementation of non-Abelian braiding impossible. Instead, our proposal of using the phonon spectrum of real materials provides a feasible platform to study non-Abelian braiding *in a single sample* by changing readily accessible experimental parameters.

More generally, the preceding discussion reflects the reason why the vast majority of condensed matter researchers are concerned with real materials: while metamaterials are a nice route to explore new properties, the real interest is ultimately in describing real physical situations, that is, real materials, with potentially practical applications. To clarify this point further with an additional example: electrical circuits can directly emulate *any* Hamiltonian, but this does not mean that once an electrical circuit emulating a particular Hamiltonian has been designed, there is no further interest in translating this to real materials. The Reviewer surely agrees that the whole field concerned with finding topological materials is very relevant, while these effects can probably be more simply achieved in toy models realised in electrical circuits. Overall, we find the argument that certain designer systems can be more easily tuned and are therefore more interesting than real materials puzzling.

In addition, applying 7% stress is not easy for experimentalists. In this context, this paper does not have significant novelty in the context of the current understanding of the non-Abelian topological physics.

Answer: Firstly, we would like to highlight here that stress is a measure of internal force while strain is the correct physical quantity measuring the deformation.

But more generally, we are a bit puzzled by this comment. The Reviewer did not raise this point in their original review, so we find it strange that they raise it now. This is even more remarkable given that the other two Reviewers did raise this point in the first round of reviews, and that we thoroughly addressed it to the full satisfaction of both Reviewers 2 and 3, which recommend publication (see detailed discussion in the original round of reviews).

3) As stated by the authors, the Raman spectrum does not show an evidence of the non-Abelian braiding but only shows a band structure modification. It is not enough as a smoking gun experiment of the non-Abelian braiding. The authors raise an example of ARPES, where only the surface state is observed. However, this is not the case because the emergence of the surface state is a strong evidence of a topological phase. On the other hand, there is no way to confirm the non-Abelian nature of the braiding experimentally. This is not enough for the publication of this manuscript in *Nature Communications*. The authors should propose a direct method to experimentally observe non-Abelian algebra of the braiding.

Answer: Here we again have to strongly disagree with the Reviewer.

First of all, the Reviewer states that surface states are a strong indication of topological bulk quantities, and contrasts it with our proposal of using the Raman spectrum evolution to track non-Abelian braiding, which the Reviewer claims it does not provide a comparable degree of evidence. This difference is false: surface states can have a non-topological origin and instead arise because of impurities, surface reconstruction, or lattice mismatch, and non-topological surface states are particularly prominent in narrow gap materials. Even for silicon, trivial surface states can emerge throughout the entire Brillouin zone, as shown in Fig. 14 and Fig. 15 in [*Phys. Rev. B* 25, 3975 (1982)]. This implies that in *all* ARPES studies, the only way to confirm the topological origin of surface states is by comparison to theory, usually in the form of first principles density functional theory calculations.

Given this, we argue that our proposed experimental signature is of an equivalent quality: the band inversion tracked by the evolution of the Raman signal indicates a change in irrep and thus a transfer of non-Abelian charge, as confirmed by the theoretical calculations. Therefore, an experiment tracking the motion of the Raman peaks associated with the modes we identify will enable experimental colleagues to track the braiding process. We actually view our proposed experiment, which directly addresses the bulk topology of the system as opposed to the derived boundary states, as a rather nice result as there are not many setups addressing bulk topology experimentally.

We should add that it is possible to probe the full phonon spectrum of materials using inelastic neutron scattering,¹⁻³ and in principle this would give access to all crossing points in the spectrum. Other techniques include inelastic X-ray scattering⁴ and high resolution electron energy loss spectroscopy,⁵ which can also be used to observe the topological band crossing points directly (and we note that these three alternative techniques had already been proposed in our original manuscript). However, we still emphasize that the Raman signature highlighted in our manuscript is a neat way of tracking non-Abelian braiding using a rather accessible and well-established technique, in contrast to neutron scattering which requires large-scale dedicated facilities.

4) The characteristic feature of the edge state for the non-Abelian topological system is still not clear enough. It seems that the edge state only reflects the Zak phase, which is irrelevant to the patched Euler number. How can we detect the non-Abelian nature from the evolution of the edge state?

Answer: The bulk-boundary correspondence, as we explained in our response to the first round of reviews, is still the subject of intensive investigation for multi-gap topology, and therefore beyond the scope of the present work. As described in the manuscript, we therefore focus on the bulk features, and the main message is that phonons provide a new and feasible platform for non-Abelian braiding.

We nonetheless qualitatively explore the edge states by considering the Zak phase, which is the current state-of-the-art given the lack of a general bulk-boundary correspondence theory. The statement by the Referee that the Zak phase has nothing to do with the patch Euler class is incorrect: we show that the Zak phase can be sufficient under certain circumstances, for example, for the three-band braiding in Group 1.

We would also like to emphasize that non-Abelian braiding occurs in the bulk states of band structures,

and this is indeed an advantage of our approach because the bulk states provide a straightforward platform compared to the real space braiding (e.g. for the Majorana zero modes) that *is* constrained to boundary states.

5) It is hard to see the braiding structure from Fig.7. In the braiding, the number of the strings should not change. On the other hand, the number of red spots changes. It looks like not a braiding but a merging process of the red spots.

Answer: We would like to refer here to the caption of this figure. First of all, as stated, the C_6 symmetry is broken, leading to the splitting of the double point at Γ , this is why one braid gives rise to two braids (red). As a second step, the braiding is achieved, as displayed in the attached figure that explains the transfer of charge (frame charge -1 , or equivalently Euler class 1) through the braiding. Finally, we restore C_6 , leading to the merging of the two blue braids into a single braid at Γ . Hence, we respectfully disagree that this was not clear. However, to explicitly show the braiding process, we have updated the original figure with the attached figure to further clarify this point, as follows,

Figure 1. Schematic braiding process in the vicinity of Γ with broken C_6 symmetry, leading to the transfer of charge (frame charge -1 , equivalently Euler class 1) from gap Δ_{11} (red triangles) to gap Δ_{10} (blue circles). The successive panels correspond to the braiding of Fig. 7(b) with $\varepsilon < t < 1 - \varepsilon$. The panels (b), (c) and (d) have the same positions of nodes, while the Dirac strings are moved around and the charges are flipped accordingly.

Reviewer #2 (Remarks to the Author):

I thank the authors for their significant efforts to answer my questions and improve the manuscript. I think the authors answered my questions clearly. I recommend the publication of the revised manuscript.

Answer: We would like to thank the reviewer for their work in reviewing our manuscript, for their very positive evaluation, and for their detailed comments in the previous round.

Reviewer #3 (Remarks to the Author):

The authors have made extensive revisions to address all the reviewers' comments, in particular how the evolution of Raman spectrum is linked to the braiding process. I am happy to recommend the publication of the work to *Nature Communication*.

Answer: We would like to thank the reviewer for their work in reviewing our manuscript and for their high evaluation of our work.

References

1. Zhang, T. T. *et al.* Phononic helical nodal lines with \mathcal{PT} protection in mob_2 . *Phys. Rev. Lett.* **123**, 245302 (2019).
2. He, X. *et al.* Anharmonic eigenvectors and acoustic phonon disappearance in quantum paraelectric SrTiO_3 . *Phys. Rev. Lett.* **124**, 145901 (2020).
3. Choudhury, N., Walter, E. J., Kolesnikov, A. I. & Loong, C.-K. Large phonon band gap in SrTiO_3 and the vibrational signatures of ferroelectricity in $a\text{TiO}_3$ perovskites: First-principles lattice dynamics and inelastic neutron scattering. *Phys. Rev. B* **77**, 134111 (2008).
4. Miao, H. *et al.* Observation of double weyl phonons in parity-breaking fesi. *Phys. Rev. Lett.* **121**, 035302 (2018).
5. Jia, X. *et al.* Anomalous acoustic plasmon mode from topologically protected states. *Phys. Rev. Lett.* **119**, 136805 (2017).